# Developing a tile drainage module for Cold Regions Hydrological Model: Lessons from a farm in Southern Ontario, Canada

**Mazda Kompanizare**[*&#], **Diogo Costa**[+*], **Merrin L. Macrae**[&], **John W. Pomeroy**[*], **Richard M. Petrone**[&]

[*]**Centre for Hydrology, University of Saskatchewan, Canmore and Saskatoon, Canada**

[+]**University of Évora, Mediterranean Institute for Agriculture, Environment and Development, Portugal**

[&]**University of Waterloo, Waterloo, Canada**

[#]**Corresponding author: kompanizare.mazda@usask.ca**

## Abstract

Systematic tile drainage is used extensively in agricultural lands to remove excess water and improve crop growth; however, tiles can also transfer nutrients from farmlands to downstream surface water bodies, leading to water quality problems. Thus, there is a need to simulate the hydrological behaviour of tile drains to understand the impacts of climate or land management change on agricultural runoff. The Cold Regions Hydrological Model (CRHM) is a physically based, modular modeling system developed for cold regions. Here, a tile drainage module is developed for CRHM. A multi-variable, multi-criteria model performance evaluation strategy was deployed to examine the ability of the module to capture tile discharge under both winter and summer conditions (NSE>0.29, RSR<0.84 and PBias <20 for tile flow and water table simulations). Initial model simulations run at a 15-min interval did not satisfactorily represent

tile discharge; however, model simulations improved when the time step was lengthened to
hourly but also with the explicit representation of capillary rise for moisture interactions between
the rooting zone and groundwater, demonstrating the significance of capillary rise above the
water table in the hydrology of tile drains in loam soils. Novel aspects of this module include the
sub-daily time step, which is shorted than most existing models, and which may enable future
water quality modules to be added, and the use of field capacity and its corresponding pressure
head to provide estimates of drainable water and the thickness of the capillary fringe, rather
using than detailed soil retention curves that may not always be available. An additional novel
aspect is the demonstration that flows in some tile drain systems can be better represented and
simulated when related to shallow water table dynamics.
Keywords: tile drainage, cold regions, hydrological model, capillary fringe, drainable water,
water table fluctuations

## 1.    Introduction

Harmful algal blooms and eutrophication in large freshwater lakes surrounded by agricultural
lands are major environmental challenges in Canada and globally. The transport of nutrients,
particularly phosphorus, in runoff from agricultural fields into rivers, ponds and eventually lakes
is an important contributor to the increased frequency of algal blooms being experienced in
North America and elsewhere (Sharpley et al., 1995; Correll, 1998; Filippelli, 2002; Ruttenberg,
2005; Schindler, 2006; Quinton et al., 2010; Costa et al., 2022). Nutrient transport from
agricultural fields can occur via both surface runoff and tile drainage (Radcliffe et al., 2015), and
recent increases in the frequency and magnitude of algal blooms in Lake Erie in North America
have been attributed to tile drainage (King et al., 2015; Jarvie et al., 2017). Tile drain systems
reduce the retention time of soil water, lessening waterlogging in fields and improving both crop
growth and field trafficability for farmers (Cordeiro and Ranjan, 2012; Kokulan et al., 2019a).
However, they are also important pathways for dissolved and particulate nutrients (Kladivko et
al., 1999; Tomer et al., 2015). It has been estimated that 14% of farmlands in Canada (ICID,
2018) and 45% of fields in Southern Ontario, Canada (ICID, 2018; Kokulan, 2019) are drained
by tile systems.  In Alberta, tile drains have also been used to address salinity issues (Broughton
and Jutras, 2013). Given their importance in hydrological budgets and biogeochemical transport,
there is a need to understand the controlling mechanisms of water and nutrient export from tile
systems as an integral part of the broader, modified hydrological system. The ability to integrate
a dynamic quantification of tile drainage from fields in hydrological models can help understand
the relative importance of this human-induced process as it interplays with an array of other
phenomena, including energy and physical mass balance hydrological processes, climate change,
and the impacts of modified land management practices on runoff and nutrient export.
There are several models that can represent tile drainage at the small basin scale, such as
HYPE (Lindstrom et al., 2010; Arheimer et al., 2015), DRAINMOD (Skaggs, 1978, 1980a;
Skaggs et al., 2012), MIKE SHE (Refsgaard and Storm, 1995) and SWAT (Arnold et al., 1998;
Koch et al., 2013; Du et al., 2005; Du et al., 2006; Green et al., 2006; Kiesel et al., 2010).  These
models include conceptual components for many key hydrological processes, but research shows
that they have been primarily designed and tested for temperate regions (Costa et al., 2020a).  In
Canada and other cold regions, some unique hydrological processes such as frozen soil,
snowmelt, rain on snow, and runoff over and infiltration into frozen or partially-frozen soils may
be very important (Rahman et al., 2014; Cordeiro et al., 2017; Pomeroy et al., 1998, 2007; Fang
et al., 2010, 2013). Many hydrological processes, such as the sublimation of snow, energy
balance snowmelt, and infiltration into frozen soils, are strongly affected by temperature and the
phase changes of water, which make many existing models developed for warm regions less
appropriate for regions with cold seasons (Pomeroy et al., 2007; Pomeroy et al., 2013; Pomeroy
et al., 2016; Fang et al., 2010, 2013). Even for temperate regions, the representation of cold
season processes is often underrepresented in models (Costa et al., 2020a).
Since the use of tile drainage is becoming popular in many cold regions, it has become
important to integrate such human-induced process in specialized hydrological modelling tools
for these regions, such as the Cold Regions Hydrological Modelling platform (CRHM, Pomeroy
et al., 2007; 2013; 2022). CRHM was initially developed in 1998 to assemble and explore the
hydrological understanding developed from a series of research basins spanning Canada and
elsewhere into a flexible, modular, object-oriented, multiphysics platform for simulating
hydrological processes and basin response in cold regions (Pomeroy et al., 2007; 2022).  The
modular CRHM platform allows for multiple representations of forcing data interpolation and
extrapolation, hydrological model spatial and physical process structure and parameter values.
Many existing models typically operate at default daily or monthly time intervals, which is
inadequate for the prediction of many short-duration "flashy" hydraulic responses often observed
in tiles (Pluer et al., 2020; Vivekananthan, 2019; Vivekananthan et al., 2019; Lam et al., 2016a,
2016b; Macrae et al., 2019). Indeed, the ability to simulate shorter time intervals (e.g., hourly)
facilitates the ability to capture both the rising and falling limbs of tile flow hydrographs, as well
as the magnitude of peak flows, both of which are important to tile drain chemistry and export
(Rozemeijer et al., 2016; Williams et al., 2015, 2016; Macrae et al., 2019).

Hydrological process models such as DRAINMOD, MIKE SHE and SWAT use a

combination of empirical and physically based formulations for the simulation of tile flow
derived by Hooghoudt (1940), Kirkham (1957), van Schilfgaarde (1974), Bouwer and van
Schilfgaarde (1963) and Skaggs et al., (1978). Such formulations contemplate both cases where
the water table is below and above the ground surface (Kirkham, 1957). In contrast, simulations
of tile drainage in other models such as HYPE use empirically derived recession curves
(Eckersten et al., 1994) to simulate tile flow and soil hydrological storage (typically represented
as water table). In cases where there is a need for more focus on soil matrix hydrology and less
need for understanding hydrological processes at the catchment scale and the relative
contribution of tiles (and its interplay), modellers tend to use specialised porous-media PDE-
based (partial differential equation-based) numerical models such as HYDRUS (Simunek et al.,
2011) and MACRO (Larsbo and Jarvis, 2003).

The amount of water transported by tiles depends on soil moisture dynamics and the

positioning of the water table, which are in turn affected by many factors, including soil type,
surface topography and morphology, as well as the local climate and the hydrological
characteristics of the field (Frey et al. 2016; Klaiber et al., 2020; Coelho et al., 2012; King et al.,
2015). Thus, to provide reliable estimations of water loss from farmland via surface runoff and
tile flow, models must be able to predict soil moisture storage and the water table elevation
accurately (Brockley, 1976; Rozemeijer et al., 2016; Javani-Jouni et al., 2018).  Many studies
have shown that in some soil types, including silty loam and clay loam soils, the drainable water
is less than expected based on the effective porosity (*e.g.*, Skeggs et al., 1978; Raats and
Gardner, 1974). Raats and Gardner (1974) have argued that the calculation of drainable porosity
requires knowledge of water table elevation and the distribution of soil moisture above the water
table. Skaggs et al. (1978) added that the calculation of drainable porosity should consider "the
unsaturated zone drained to equilibrium with the water table". However, because the soil column
is often composed of different soil layers with varying physical characteristics, drainable
porosity varies with evapotranspiration rate, soil water dynamics and the depth of saturated water
(Logsdon et al., 2010; Moriasi et al., 2013). In a sandy loam soil, Lam et al. (2016a, 2016b)
demonstrated that tile drainage was not initiated until soil was at or above field capacity.
Williams et al. (2019) observed in the American Midwest that tile drainage was not initiated until
the field storage capacity had been exceeded. It has also been shown that despite the presence of
tile drains, the soil above the tile may not always drain all gravitational water following a
rainfall/snowmelt event and the soil may remain at or above field capacity (Skaggs et al., 1978;
Lam et al., 2016a). Therefore, the soil drainable water content may be considerably smaller than
the storage capacity. This is related to matric potential within the vadose zone, which is driven
by the soil characteristics but can also be due to the development of a capillary fringe that
reduces the rate of vertical percolation through the unsaturated zone, reducing tile flow (Youngs,
2012). Despite this evidence, some saturated flow models that simulate tile flow overlook the
effect of capillary rise and over-estimate the soil drainable water. Other models that represent
unsaturated flow (i.e., HYDRUS 3D, Simunek et al., 2011) using Richard's Equation (Richards,
1931) capture the effect of capillary rise and saturation-pressure variation within the soil profile
and assess the soil drainable water more accurately. Although the effect of capillary rise is
considered in DRAINMOD through the concept of drainable porosity (represented as a "water
yield") (Skaggs, 1980b), and is calculated for layered soil profiles (Badr,1978), it requires
detailed information surrounding the soil water characteristic curve (Skaggs, 1980b). Although it
is indeed optimal to use soil-specific water characteristic curves, Twarakawi et al. (2009) found
that it was possible to employ average representative values from the soil water characteristic
curve to represent soil drainable water where a soil-specific curve was not available.  They found
in this case that the model performance was reduced.

In this study, a new Tile Drainage Module (TDM) was developed and incorporated

within the physically based, modular Cold Regions Hydrological Modelling (CRHM) platform
(Pomeroy et al., 2022) to enable hydrological simulations in tile-drained farm fields in cold
agricultural regions. As a first iteration, the new module was developed for a field with sloping
ground and loam soil with imperfect drainage. Such landscapes are common in the Great Lakes
Region (e.g., Michigan and Vermont, USA and Ontario, Canada) and tile drainage in such
landscapes has not been as widely studied as it has been in clay-dominated soil.  In this module,
considerations were explicitly included for the effects of capillary rise and annual groundwater
water table fluctuations on drainable soil water storage. The use of field capacity and
groundwater/soil water elevation head (Twarakawi et al., 2009) to modulate soil drainable water
across the soil profile, including the capillary fringe region, is an innovative aspect of the model
that has been demonstrated to circumvent the need for water characteristic curves. The
development of this physically based module provides insight into hydrological processes in tile
drainage from sloping landscapes with imperfect drainage, which are increasingly being
artificially drained.

## 2.    Materials and Methods

*2.1    Study area*

The study site is a ~10 ha farm field located near Londesborough, Ontario at UTM 17T 466689m

E, 4832203m N, shown as LON in Fig. 1a. Mean annual precipitation recorded in this region is

1247 mm (ECCC, 2020). Mean air temperature is 7.2 $^{\circ}$C, with annual maxima in July (25.9 $^{\circ}$C)

and minima in January (-10.2 $^{\circ}$C), (ECCC, 2020). Soil texture has been identified as Perth clay

loam (Gr. Br. Luvisolic), with a slope between 0.2 and 3.5%. The field is systematically drained

with a tile depth of 0.9 m and a spacing of 14 m (laterals). The tile network collects infiltrated

water from about 75% of the field (~ 7.6 ha) but may also receive lateral groundwater flow from

neighbouring fields. Water yields from the tile drain laterals (10 cm diameter) are discharged via

a common tile outlet (main, 15 cm diameter) below ground. Surface runoff from the field is

directed toward a common outlet on the surface using plywood berms installed along the field

edge (see van Esbroeck et al., 2016). The tile and surface runoff outlets do not join into a

common outlet and are fully separated from one another, even during surface ponding events.

The field is a corn-soy-winter wheat rotation with cover drops and rotational conservation till

(shallow vertical tillage every three years). Additional details related to farming practices are

provided in Plach et al. (2019), soil characteristics are provided in Plach et al. (2018a) and Plach

et al. (2018b) and equipment and monitoring are provided in van Esbroeck et al., (2016). The

outlets for both surface and tile flow are located at the edge of the field and drain into an adjacent

field (Fig. 1b). Water tends to accumulate in a topographic low in the field, in front of the field

outlet during snowmelt or high-intensity rainfall events, presumably due to either surface runoff

or return flow (see ponded area, Fig. 1b). However, surface water or elevated soil moisture

conditions are not observed in this topographic low during smaller events or dry periods of the

year, suggesting that this saturated ponding is not in a perennial groundwater discharge zone.
Although surface ponding is observed in the topographic depression within the field, water
discharges freely at the opposite end of the culvert, facilitating the measurement of flow.

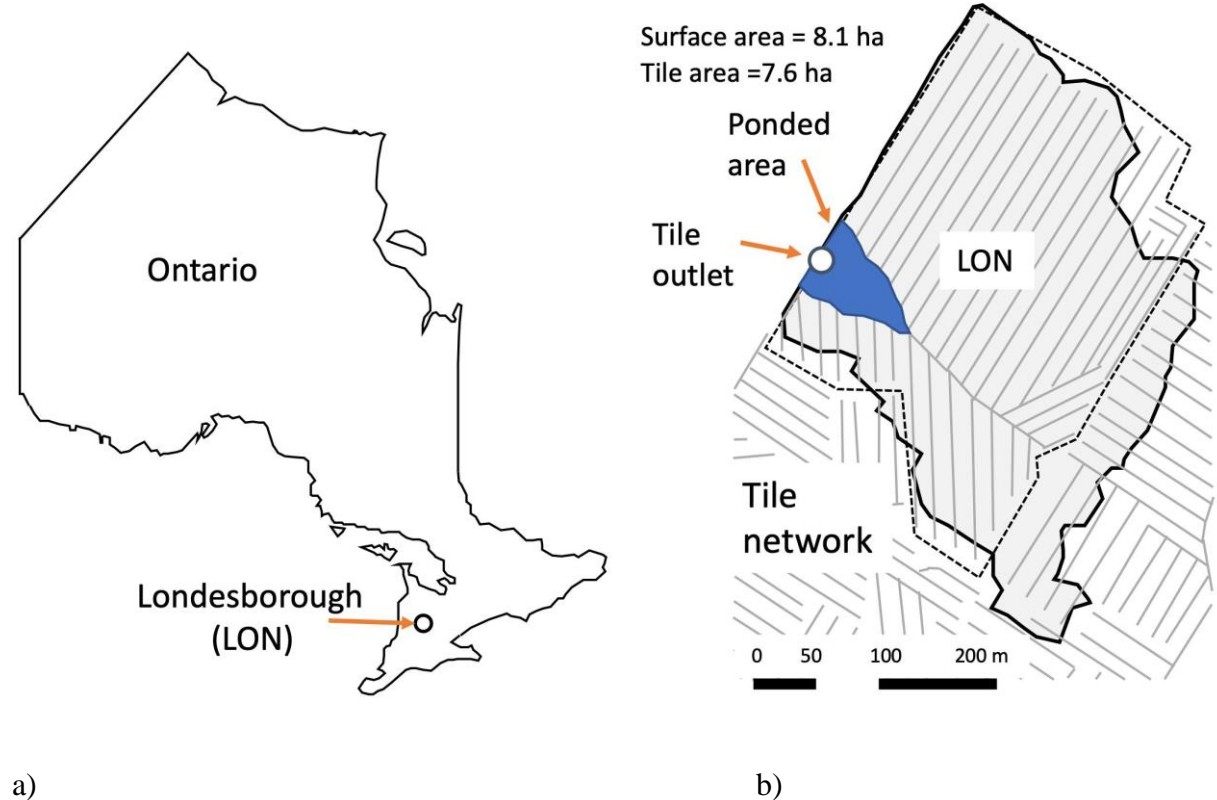


a)                                                    b)
b)    Figure 1. (a) Location of the study area in South of Ontario and the (b) Londesborough (LON) farm with its tile network.


*2.2    CRHM: The modelling platform*
The modular CRHM platform includes options for empirical and physically based calculations of
precipitation phase, snow redistribution by wind, snow interception, sublimation, sub-canopy
radiation, snowmelt, infiltration into frozen and unfrozen soils, hillslope water movement, actual
evapotranspiration, wetland fill and spill, soil water movement, groundwater flow and
streamflow (Pomeroy et al., 2007; 2022).  Where appropriate, it calculates runoff from rainfall
and snowmelt as generated by infiltration excess and/or saturated overland flow, flow over
partially frozen soils, detention flow, shallow subsurface flow, preferential flow through
macropores and groundwater flow. Water quality can also be simulated in CRHM (Costa et al.,
2021).  Modules of a CRHM model can be customized to basin setup, such as delineating and
discretizing the basin, conditioning observations for extrapolation and interpolation in the basin,
or are process-support algorithms such as for estimating longwave radiation, complex terrain
wind flow, or albedo dynamics, but most modules commonly address hydrological processes
such as evapotranspiration, infiltration, snowmelt, and streamflow discharge. CRHM discretizes
basins into hydrological response units (HRU) for mass and energy balance calculations, each
with unique process representations, parameters, and position along flow pathways in the basin.
HRU are connected by blowing snow, surface, subsurface and groundwater flow and together
generate streamflow which is routed to the basin outlet.  The size of TDM HRUs is flexible and
can be as small as the size of a single tile pipe (e.g., 1 m) times the pipe spacing (which was 14
m in our case study region), and as large as entire tile networks within a given farm or study
area. CRHM does not require a stream within a modelled basin. The feature allows CRHM to
model the hydrology of cold regions dominated by storage and episodic runoff, such as
agricultural fields.

Although CRHM has the capability to represent many hydrological and thermodynamic

processes, not all processes need/must be represented in all situations. The modular design of the
CRHM platform enables the user to activate or inactive specific processes to optimize the model
for a particular situation. This is a modelling approach that enables testing different modelling
hypotheses and has been pioneered by CRHM and other models, which has inspired a range of
hydrological (e.g., SUMMA, Clark et al., 2015a, 2015b), hydrodynamic (e.g., mizuRoute,
Mizukami et al., 2015) and biogeochemical (e.g., OpenWQ, Costa et al., 2023a, 2023b)
modelling tools. For example, in the current study, blowing snow was not employed in CRHM as
it does not appear to be significant at the study site (periodic snow surveys showed relatively
uniform snow cover). Preferential flow into tile drains was not developed for the current
simulation as although it is a key process in clay loam soil, as it does not appear to be a
significant driver of preferential flow into tile drains in coarse textured soil (Pluer et al., 2020;
Macrae et al., 2019). Freeze-thaw processes in soil were also not employed here as there is very
little seasonal soil frost in the temperate Great Lakes region due to the persistent snow cover, and
where soil frost occurs, it is restricted to brief periods and shallow depths (above 10 cm depth)
(Macrae unpublished data).

*2.3    Observations and input data for the model*
Tile flow, water table elevation (water table elevation head) and surface flow were measured at
the site between Oct. 2011 and Sept. 2018 at 15-minute intervals. It was not possible to install
more than one measuring station for water table elevation and soil moisture at the site due to
farming activity; consequently, water table elevation head and soil moisture were measured at
the approximate midpoint of the field at the edge-of-field. Both tile flow rates and surface runoff
were determined using simultaneous measurements of flow velocity and water depths in each of
the pipes at the edge-of-field using Hach Flo-tote sensors and an FL900 data logger  (Onset Ltd.)
(Table A1, Appendix A). Continuous measurements of velocity were included due to the
potential for impeded drainage under very wet conditions or caused by the accumulation of snow
and ice around the surface culvert in winter. An additional barometrically-corrected pressure
transducer (U20, Onset Ltd.) (Table A1) was also used for periods when the flow sensors did not
function using a rating curve developed from the depth-velocity sensors. The water table
elevation was measured using a barometric pressure-corrected pressure transducer (U20, Onset
Ltd.).

Air temperature, wind speed, air relative humidity, incoming solar irradiance and rainfall

were also measured at the site at 15-minute intervals and used to force the model. Variable
names and their symbols in CRHM are listed in Appendix B. The air temperature, wind speed
and incoming solar radiance measurements were collected 1 m above ground using a
Temperature Smart Sensor S-THB-M002, Wind Smart Sensor Set S-WSET-M002 and a Solar
Radiation Sensor (Table A1). Rainfall and relative humidity were measured via a tipping bucket
rain gauge (Table A1) and an RH Smart Sensor (Table A1). These observations were
continuously recorded throughout the study period, except for brief periods of instrument failure
and maintenance, when data from nearby stations (Table T1, Supplementary Material) was
substituted using the double mass analysis method (Searcy and Hardison, 1960).

Although rainfall was recorded continuously at the field site, snowfall data was not.

Snowfall data was obtained from nearby stations (Wroxeter-Davis and Wroxeter, Environment
Canada, 2021), located 31.7 km from the field site. Periodic snow surveys done at the site
throughout the study period found that data from the nearby stations was a close approximation
of snow at the field site (Plach et al., 2019). Hourly snowfall observations from Wroxeter-
Geonor were used for the period between 2015 and 2018, whereas daily data from the Wroxeter-
Geonor were used for the 2011 to 2014 period, reconstructed to hourly snowfall time series
based on the method presented by Waichler and Wigmosta (2003).

*2.4    Development of the new tile module*
A Tile Drainage Module (TDM) was developed within CRHM (Figures 2, 3) with the goal of
adding the ability to simulate tile flow and the resulting saturated storage (water table) at an
hourly time step. CRHM was forced with hourly precipitation, air temperature, solar radiation,
wind speed and relative humidity to calculate hydrological states and fluxes in HRUs and the
basin. The model requires parameterizations that specify the hydraulic and hydrological
properties of the soil, including its thickness, saturated hydraulic conductivity (K), and surface
cover. CRHM calculates water storage and fluxes between HRUs, as well as vertical fluxes
amongst different hydrological compartments (within each HRU) that include snow,
depressional storage, different soil layers, and groundwater.

Using the simulation of soil moisture (including both saturated and unsaturated soil

moisture) performed by the original CRHM "*Soil*" module, TDM calculates the dynamic tile
flow rate that, in turn, feeds back to soil moisture at each time step. The presence of a capillary
fringe (sometimes referred to as the tension-saturated zone within the soil profile) and its effects
are considered by limiting the amount of drainable soil water. TDM uses site-specific
information regarding the tile network, such as tile depth, diameter and spacing. Information
regarding site-specific details regarding tile depth, diameter and spacing may be obtained
directly from landowners or can be estimated based on standard design and installation
guidelines for the region. This information was used to set up the model together with
parameterization to translate the hydrological effects of the soil capillary fringe (CF), if present,
through two variables, CF thickness and CF drainable water (discussed in Section 2.5, Figures 2,
3). These two variables are used to limit the fraction of the soil moisture that can freely drain to
the tiles.

### 2.4.1   Soil moisture and water table elevation

The TDM uses the water quality soil module or soil module (*WQ_soil* or *Soil*), which divides the

soil column into two layers: a recharge layer where evapotranspiration and root uptake generally

take place and a deeper layer that connects to the groundwater system. Since CRHM's state

variable for soil moisture is soil water storage volume (Fig. 2), the model results were converted

into water table elevation above the semi-permeable layer (Table B1, Appendix B; see Fig. 2b

for comparison with water table observations) by dividing volumetric soil moisture content

(Table B1) by soil porosity (Table B1) for the cases with no capillary fringe above the water

table. Additional steps were taken for periods when a capillary fringe developed (discussed

below).

### 2.4.2   Capillary fringe and drainable water

Soil moisture in the capillary fringe is equal to the average volumetric water content at capillary

fringe ($\theta_C$) which is usually greater than the field capacity ($\theta_{fc}$) (Bleam, 2017, Sect. 2.4).

Therefore, while the positioning of the capillary fringe responds dynamically to the matric

potential, the saturation profile within the capillary fringe remains constant, as well as its

thickness because it only depends on the pressure head (capillary forces) that are related to the

grain size distribution and field capacity ($h_{fc}$) as introduced by Twarakawi et al. (2009).

Therefore, the drainable water in the capillary fringe becomes the difference between saturation

($\theta_s$), computed dynamically in CRHM, and $\theta_C$, which corresponds to the water held by capillary

forces at the capillary fringe moisture content (Fig. 2). Accordingly, Fig. 2 shows the schematic

soil characteristic curve for the three water level conditions contemplated in the model.

1. *Condition 1* is when the water table is at the surface and the soil is completely saturated
(matric potential = 0);
2. *Condition 2* is when the water table drops but the upper boundary of the capillary fringe
is at the soil surface; and
3. *Condition 3* is when the water table drops further, and the upper boundary of the
capillary fringe drops beneath the surface.
In essence, the soil is completely saturated ($\theta_s$) in *Condition 1*. Between *Conditions 1* and *2*, the
capillary fringe occupies the entire soil column above the water level; thus, it can only release
the volume of water corresponding to $\theta_s$-$\theta_C$ or $\varphi_c$ (dimensionless). Between *Conditions 2* and *3*,
two layers with distinct hydraulic characteristics develop: (1) the top one at $\theta_{fc}$ that releases
water up to $\theta_C$-$\theta_{fc}$, and (2) the lower one that corresponds to the capillary fringe and can release
up to the volume of water corresponding to $\theta_s$-$\theta_C$ or $\varphi_c$.

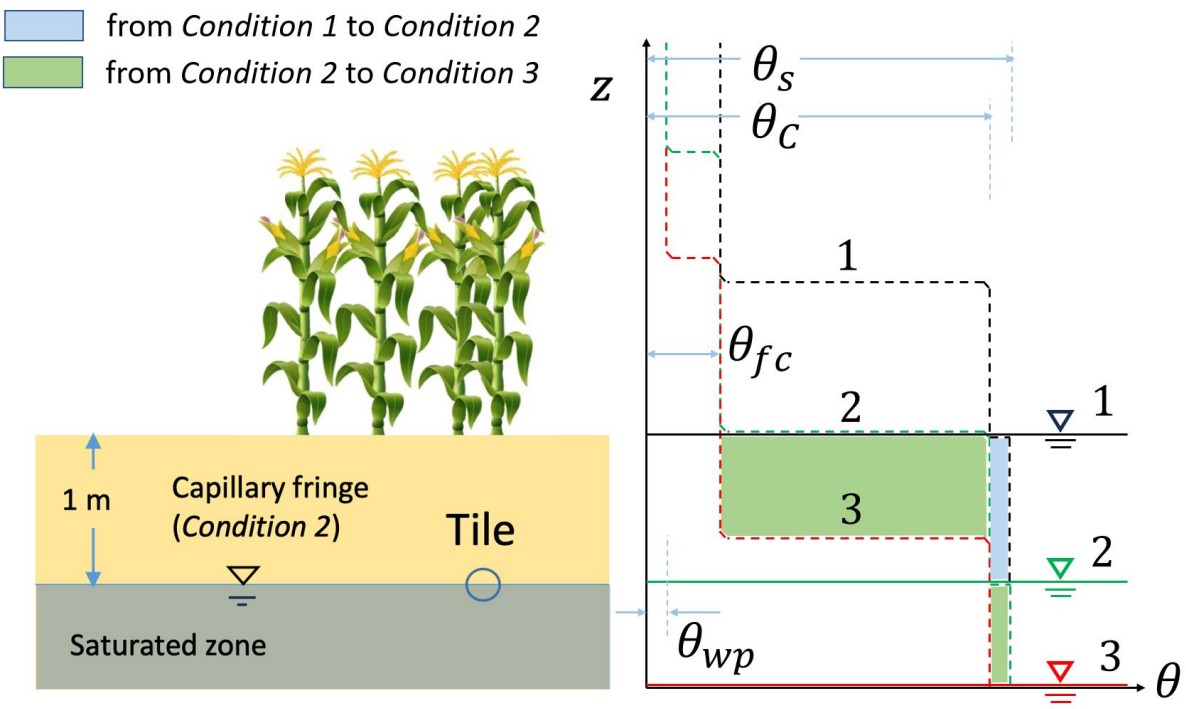


Figure 2. Schematic representation of the capillary fringe above the water table assuming a 1-m thickness (for demonstration
purposes). The soil characteristic curves are shown for the three water level conditions considered: water level at the (1) surface,
(2) intermediate depth, and (3) deeper depth. Two transitional drops can be seen in the characteristic curves, one from saturation
($\theta_s$) to capillary fringe water content ($\theta_C$) (between *Conditions 1* and *2*) and one from $\theta_C$ to field capacity ($\theta_{fc}$) (between
*Conditions 2* and *3*). The coloured areas (green and blue) of the right panel correspond to the amount of water that can be
released between *Conditions 1* and *2* (blue) and between *Conditions 2* and *3* (green).


*2.4.3   Tile flow calculation*
A modified version of the Hooghoudt equation was used to calculate tile flow (Smedema et al.,
2004), which presumes no surface ponding, an assumption that generally holds at the study site
(Eq. 1), where water ponds only during very wet periods and on a small portion of the study site
(see Fig. 1b). Hooghoudt's equation (Hooghoudt, 1940) is a steady state, physically based
equation for saturated flow toward the tile drain. Flow estimates are provided based on the
hydraulic conductivity of the soil and water table elevation above the tile pipe.  It allows
different saturated hydraulic conductivities for the layers above (AL) and below (BL) the tile
(Fig. S1). At the study site, soil surveys have reported almost the same soil type (Loam) down to
the depth of 90 cm (*e.g.,* Van Esbroeck et al., 2016; Plach et al., 2018b), which was
parameterized in the model set up as,

$$q = \frac{8 \times K_2 \times d \times h}{L^2} + \frac{4 \times K_1 \times h^2}{L^2} ,$$
(1)


where $K_1$ and $K_2$ are respectively the saturated hydraulic conductivity in the upper and lower
layers in mm h$^{-1}$; $L$ is the tile spacing in mm; $h$ is the water table elevation above the tile in mm,
$d$ is the lower layer thickness in mm (Fig. S1), and $q$ is the predicted tile flow in mm h$^{-1}$. The
only variable that is dynamically updated by CRHM is $h$. Equation (1) is used to estimate the tile
flow.

*2.4.4    Calculation of the effect of tile flow on soil moisture and water levels*
The simulated tile flows (see Sect. 2.3.3) are subtracted from the soil moisture. To calculate
saturated storage (water table or groundwater elevation head level) from soil moisture calculated
by the model, a threshold soil moisture content ($sm_t$) is defined, which consists of drainable
water in the soil ($\varphi_c$) when the upper boundary of the capillary fringe is at the surface (*Condition*
*2*, Fig. 2) and was calculated as:

$sm_t = sm_{max} - (C_t \times \varphi_c)$ ,                                                         (2)

where $sm_{max}$ is the maximum soil moisture and $C_t$ is the capillary fringe thickness in mm.
However, since the hydrological conditions of the soil are markedly different between the two
transitional situations described in Sect. 2.3.2 and Fig. 2 (*Condition 1* to *2* and *Condition 2* to *3*),
a step function was deployed for determination of the water table elevation:

$$WT = \begin{cases} \dfrac{sm_t - \left(C_t \times \left((\varphi_s - \varphi_c) + \theta_{fc}\right)\right)}{\varphi_s + \theta_{fc}} + \dfrac{sm - sm_t}{\varphi_c} & , if\ between\ Conditions\ 1\ and\ 2 \\[4mm] \dfrac{sm_{max}}{\varphi_s + \theta_{fc}} - \left(\left(\dfrac{sm_t - sm}{\varphi_s}\right) + C_t\right) & , if\ between\ Conditions\ 2\ and\ 3 \end{cases}$$        (3)

where *WT* is water table elevation (or soil saturated storage, SSS) in mm from the bottom of the
soil, and *sm* is soil moisture (both saturated and unsaturated storage) in the given time step in
mm. Equation (3) is determined based on soil moisture curves in Fig. 2 and water level
*Conditions 1-3* discussed in Sect. 2.3.2. In Fig. 2, the first and second parts of Eq. (3), which
refer to *Conditions 1* to *2* and *2* to *3*, respectively, correspond to the volumes of soil water
highlighted in "blue" and "green."

*2.4.5   Lower semi-permeable soil layer and periodicity in annual groundwater levels*
This model application focused on the study site field without including other adjacent areas.
This was possible because years of field monitoring at this site have demonstrated that there is no
observable surface flow into the site from adjacent farms. The tile network is restricted to the
field and is not connected to tile drains or surface inlets in adjacent fields. However, field soil
water table observations show evidence of annual groundwater level periodicity/fluctuation (Rust
et al., 2019) that are sinusoidal in nature and cannot be neglected. Some studies predict the
annual groundwater oscillations or the annual responses of groundwater to precipitation by using
sine and cosine functions (De Ridder et al., 1974; Malzone et al., 2016; Qi et al., 2018). De
Ridder et al. (1974) studied the design of the drainage systems and described the seasonal
groundwater fluctuations observed in wells using sinusoidal curves. Malzone et al. (2016) used a
sine function to predict annual groundwater fluctuations in the hyporheic zone. Qi et al. (2018)
and Rust et al (2019) used a cross-wavelet transform, consisting of the superposition of sine and
cosine curves, to predict shallow groundwater response to precipitation at the basin scale. This
approach was used in this application to simulate annual fluctuations in groundwater water table,
in Eq. (4), over a period of 1 year, with minimums around the middle of the growing season
(mid-July), and maximums in the cold season (early February). This translates into the lower
matric potential during the growing season, coinciding with soil moisture depletion, and then
during the non-growing season, an elevated matric potential coinciding with an increase in the
soil moisture, consistent with field observations. Thus, a sine function representing the annual
fluctuations in percolation rate from soil to groundwater ($G_{y,i}$) layers in CRHM, through the
lower soil semi-permeable layer (in mm hr$^{-1}$) is defined as:

$$G_{y,i} = \left[ A \times sin\left( \frac{(T_s - D_d \times 24) \times 360}{24 \times 365.25} \right) - B \right] \times f_{y,i} \tag{4}$$

where $T_s$ is the time step number, $D_d$ is a time delay in days, $A$ is the amplitude of the water table
(WT) fluctuation, and $B$ is an intercept factor. $f_{y,i}$ is a seasonal factor. The sine function
coefficient ($D_d$, $A$, and $B$) and seasonal factor were adjusted for the whole period and for each
year through model verification and shown in Table 1. Appendix C provides more details on the
implementation of Eq. (4).

*2.5    Model application and multi-variable, multi-metric validation*
The study site is a relatively small field, and 2 HRUs were sufficient to capture its hydrological
dynamics in CRHM. The HRUs represent (1) the area immediately upstream of the outlet where
surface ponding occurs (depression storage); and (2) the remaining field (Fig. 3). The maximum
ponding capacity of HRU 1 was estimated using the spatially distributed hydrodynamic model
FLUXOS-OVERFLOW (Costa et al., 2016, 2020b). The CRHM model with its new TDM
module were set up using the information described in Table 1. Soil textures at the LON site
measured in a 25 m grid across three soil depths (0-25 cm, 25-50 cm, and 50-100 cm) averaged

412 29% sand, 48% silt, and 23% clay (Ontario Ministry of Agriculture, Food and Rural Affairs Soil

413 Team, unpublished data). This soil grain size distribution corresponds with a soil saturated

414 hydraulic conductivity of ~ 0.56 cm h$^{-1}$ (~$10^{-2.5}$) (Garcia-Gutierrez et al., 2018), which was

415 implemented in CRHM (0.5 cm h$^{-1}$), corresponding to a field capacity of 0.04 (volumetric water

416 content) and $h_{fc}$ of ~0.8 m (Twarskawi et al., 2009, based on a drainage flux of 0.1 cm d$^{-1}$).

418 A robust multi-variable, multi-metric model evaluation strategy was deployed to verify the

419 capacity of the model to predict tile flow and its impact on the local hydrology. The outflows

420 examined were tile flow, surface flow, and water table depth. The multi-metric approach

421 contemplated five different methods, namely the Nash-Sutcliffe efficiency ($NSE$), Root-Mean-

422 Square Error (RMSE), Model Bias (Bias), Percentage Bias (PBias), and RMSE-observation

423 standard deviation ratio (RSR). See Appendix C for more details about the methodology used. It

424 is generally assumed that $NSE$>0.50, $RSR \leq 0.70$, and $PBias$ in the range of $\pm 25\%$ are

425 satisfactory for hydrological applications (Moriasi et al., 2007). Five different metrics were used

426 to evaluate model accuracy in order to describe different aspects of the discrepancies between

427 simulated and observed values. For example, Bias reveals the positive or negative general

428 deviations of simulated values from the observed values, while RMSE shows the average

429 absolute differences between them (Moriasi et al., 2007). Hourly values were used in these

430 calculations, which departs from the daily and monthly analyses typically reported for these

431 types of models. Although the hourly timestep is challenging for this sort of simulation, it is an

432 important advance forward toward more detailed, accurate, and advanced models for tile drained

433 agricultural fields. For example, Costa et al., (2021) noted that the successful extension of

434 hydrological models to water quality studies relies on their ability to operate at small time scales

in order to capture intense, short-duration storms that may have a disproportional impact on the
runoff transport of some chemical species such as phosphorus – in essence, to capture hot spots
and hot moments for flux generation.

Table 1. Key model parameters in CRHM for representation of the LON site.

| Model Parameter | Value | Unit | Source | Adjusted/Calibrated | Comment |
|---|---|---|---|---|---|
| Soil depth or Soil thickness, $T_{SL}$ | 2 | m | | No | Assumed |
| Semipermeable layer depth | 3 | m | | No | Assumed |
| Tile depth | 0.9 | m | | No | Farmer/Blueprints of the field |
| Corn root depth | 0.5 | m | | No | Online sources |
| Soil recharge zone thickness | 0.5 | m | | No | Based on the root depth |
| Tile spacing | 14 | m | | No | Farmer/Blueprints of the field |
| Soil porosity (soil drainable water) $\varphi_S$ | 0.045 | | | Yes | Adjusted |
| Saturated Hydraulic conductivity, $K$ in lower soil layer | 5 | mm h$^{-1}$ | | Yes | Adjusted |
| $K$ in upper soil layer | 5 | mm h$^{-1}$ | | Yes | Adjusted |
| Capillary fringe thickness, $T_{CF}$ | 0.8 | m | | Yes | Adjusted |
| Capillary fringe drainable water, $\varphi_C$ | 0.03 | | | Yes | Adjusted |
| Surface depression close to farm surface flow outlet (HRU2) | 35 | mm | | Yes | Calculated |
| Surface depression in rest of the field (HRU1) | 0 | mm | | No | Calculated |

| | | | | |
|---|---|---|---|---|
| Surface area of HRU1 | 79000 | m$^2$ | No | Field observations and DEM |
| Surface area of HRU2 | 1000 | m$^2$ | No | Field observation and DEM |
| Soil module name in CRHM | WQ_soil | | No | |
| Infiltration module name in CRHM | GreenAmpt | | No | |
| Soil type in GreenAmpt module | 5 | | Yes | Adjusted |
| Saturated K in GreenAmpt module | 6 | mm h$^{-1}$ | Yes | Adjusted |
| Soil wilting point | 0.025 | | Yes | Adjusted |
| $A$, in sine function | 0.025 | mm h$^{-1}$ | Yes | Adjusted |
| $B$, in sine function | -0.005 | mm h$^{-1}$ | Yes | Adjusted |
| $D_d$, in sine function | 15 | d | Yes | Adjusted |
| $f_{2012,2}$ (Seasonal factor, sine function) | 2.0 | | Yes | Adjusted |
| $f_{2015,2}$ (Seasonal factor, sine function) | 1.8 | | Yes | Adjusted |
| $f_{2016,2}$ (Seasonal factor, sine function) | 2 | | Yes | Adjusted |
| $f_{2017,2}$ (Seasonal factor, sine function) | 1.4 | | Yes | Adjusted |
| $f_{y,i}$ | 1 | | No | By default for $y = 2012\ to\ 2017$ and $i = 1, 2$ |




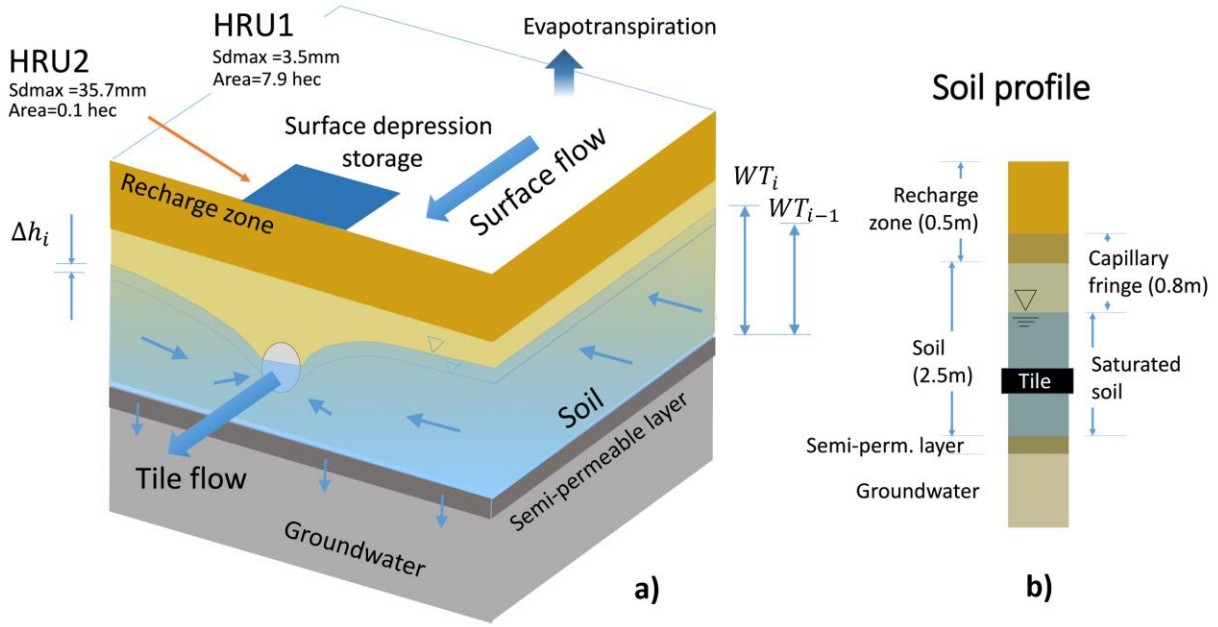


Figure 3.  a) Schematic conceptual view of the CRHM model configuration, including soil layers, water table (WT),

groundwater, and tile flow.; and b) soil profile, including the capillary fringe and its location relative to the soil and tile.

## 3.    Results

*3.1    Tile flow*

The model was able to capture most tile flow events, both in terms of the timing and magnitude

of peak flows and the most important seasonal patterns (Fig. 4). For example, the almost

complete absence of tile flow during the growing season (May to September) was captured. The

simulated flow peaks generally had a good agreement with observations, as well as the low flow

or base flows during cold periods (December-March). The ascending and descending limbs of

the response signal were also adequately predicted.

Results show that tile flows generally occurred during snowmelt events, as indicated by the

synchrony between snow water equivalent (SWE) depletion and tile flow. The maximum

snowpacks (or snow water equivalent, SWE) were markedly smaller during the winters of 2016
and 2017 when compared with those of 2013 to 2015. However, this did not necessarily translate
into lower tile flows as precipitation also occurred as rain during these seasons.  Although the
magnitude of tile peaks was not always predicted accurately, the model was able to capture the
annual trends of both an absence of tile flow during the summer months (growing season) and
the ascending and descending limbs of the tile hydrograph during events (Figure 4).

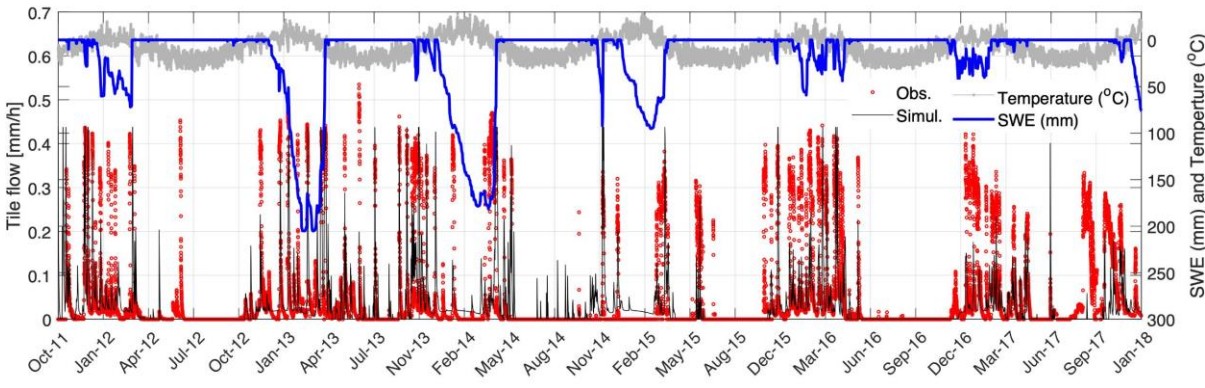


a)

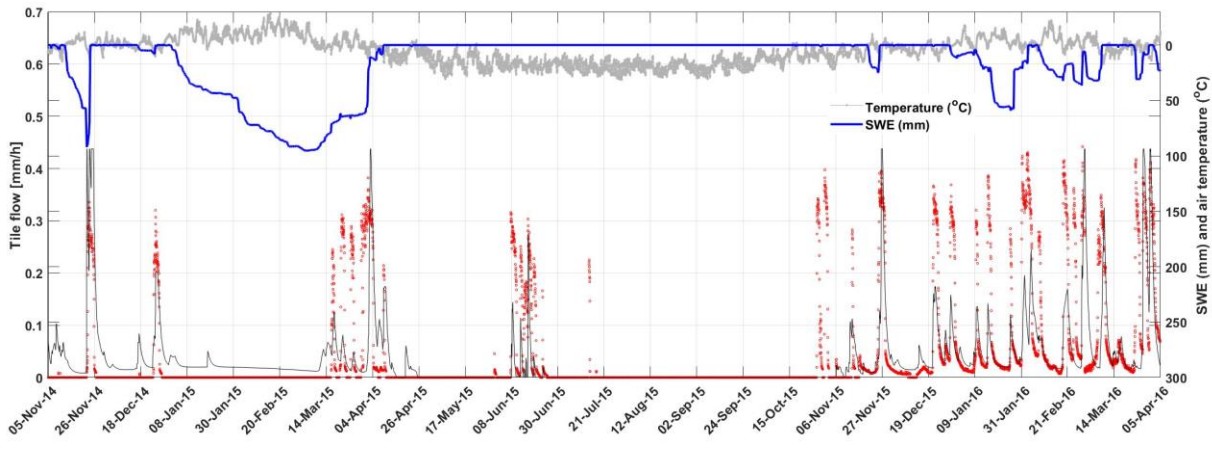


b)
Figure 4. Comparison between observed and simulated tile flows, simulated SWE (snow water equivalent), and observed air
temperature in the LON site, between October 2011 to January 2018 (a) and between November 2014 to April 2016 (b).

*3.2    Water table or soil saturated storage*
Simulated soil saturated storage and the observed water table are compared in Fig. 5, alongside
air temperature and precipitation observations. Despite the gaps in the observational record
during two periodic equipment failures, the model agrees well with observations. Above tile
drains, water table fluctuations were controlled by infiltration/recharge, tile flow, groundwater
flow, and matric potential that affect the drainable water from the capillary fringe. This caused
flashier storage responses above the tile that were captured well by the model.  In contrast, tiles
did not withdraw water from the soil layer below the tile pipe and thus did not control water table
fluctuations when levels were below the drain pipe, and tile drains did not flow during such
periods.  During the growing season, both the observed and simulated water table (or saturated
storage) dropped abruptly because of the seasonal lowering of the regional groundwater water
table. In the growing seasons of 2012, 2015 and 2016, which were dry years, large declines in
the water table and saturated storage were observed, whereas in wetter years such as 2013 and
2014, seasonal water level declines were smaller. The seasonal declines in water level during the
growing season led to a cessation in tile flow in most years (Fig. 4, 5), even following rainfall
events. For example, there was a large precipitation event (~35 mm) in the growing season of
2016 that did not produce tile flow (apparent in both model and observations).

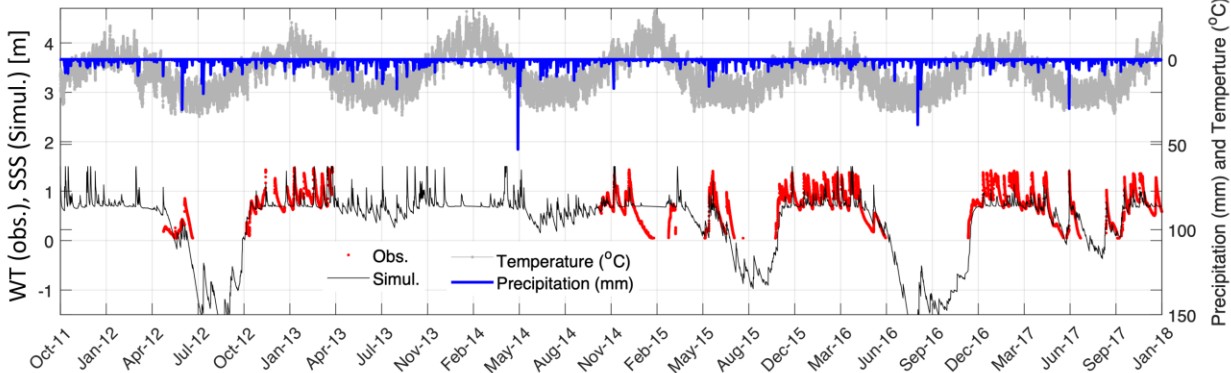


Figure 5. Time series of the simulated saturated storage and observed water table in the soil or groundwater layers of the model along with the observed temperature and precipitation. Given that tiles do not flow when the WT is below them, the WT = 0 when the water table elevation is at the depth of the tile drainpipe. In the figure, the water table is measured as the elevation above (+) or below (-) the tile pipe.

### 3.3    Surface flow and total flow

The model was not always able to capture the observed surface flow as satisfactorily as it captured tile drainage (Fig. 6a). Some possible reasons are uncertainties in the measurements of surface flow due to ponding in surface depressions on the field, which impeded the drainage of some of the surface runoff prior to when it exited the field through the culvert (see Fig. 1), or uncertainty in field estimates of SWE. However, the model performance improves considerably when both runoff and tile flow are combined (referred to as total flow, Fig. 6b). Indeed, most of the flow from the field was through tile drains (80% in 5-year average) rather than surface runoff (20% in 5-year average, Plach et al., 2019). The underestimation of both cumulative total and surface flows during 2017 and 2018 is possibly due to the removal of the blockage in the tile pipe in early 2017, which may have affected both surface and tile flow. The differences in timing of the simulated and observed surface flow for many of the main events (Figure 6) shows that there remain systematic issues in simulation of surface flow by CRHM which should be addressed in future research.

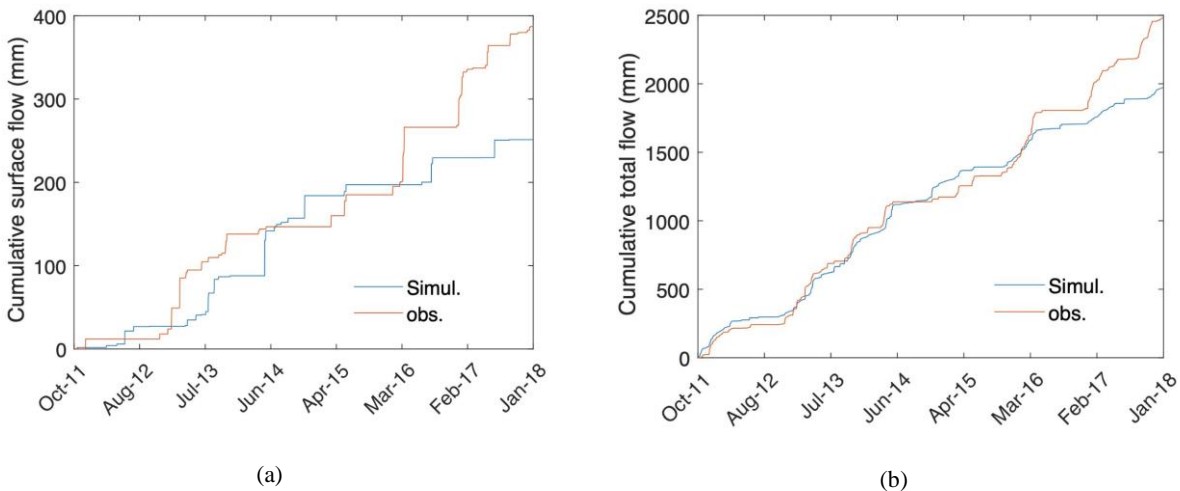

 Figure 6. Observed and simulated cumulative surface flow (a) and total flow (b).


*3.4    Overall model performance*
The model performance was calculated based on hourly data for various model outputs (Table
2). The results confirm that the model is robust in the sense that it can capture the main patterns
of tile flow, surface flow, and water table elevation. The PBias values are below 25% for most of
the fluxes and cumulative fluxes. The RSR values are also generally below 1.0. The NSE values
are positive and above 0.3 for most fluxes, except for surface flow, where the model exhibited
some difficulties. The weaker performance of the model in simulation of surface flow which is
illustrated by the NSE coefficient can be partly related to difficulties in measurement of surface
flow during flooding, ponding and freeze and thaw on the surface. The performance coefficients
were calculated for May-September (Table 2b) and October-April (Table 2c). The results shows
that surface flow biases are significantly larger and negative in May-September and are smaller
and positive during October-April. For tile flow the Biases are a bit higher in May-September
while for soil water table and total flow the biases are a bit lower in May-September. The NSEs
are more acceptable in October to April for surface flow, tile flow and total flow but the NSE for
WT is more acceptable in May-September.

Table 2. Performance coefficients for surface flow, tile flow and water table (WT/SSS), as well as total (tile + surface) flow, for the simulation period of October 2011 to January 2018. The coefficients were calculated for both hourly and daily flow rates, for the whole year (a) for May to September (b) and for October to April (c). (Green and red color show the seasonal coefficients improved and worsened, respectively, compared to their seasonal values).

a) Coefficients for whole year

| Performance coefficients | Surface flow | Tile flow | WT (m) | Total flow | |
|---|---|---|---|---|---|
| NSE[*] | -2.29 | 0.31 | 0.49 | -1.38 | Coefficients calculated for hourly flow rates (mm h$^{-1}$) |
| RMSE[^] | 0.27 | 0.08 | 0.26 | 0.30 | |
| Bias[#] | 0.54 | 0.24 | 0.14 | 0.28 | |
| PBias[$] | 21.77 | 17.91 | 10.46 | 18.63 | |
| RSR[&] | 1.82 | 0.83 | 0.71 | 1.54 | |
| NSE | -0.73 | 0.29 | 0.50 | 0.01 | Coefficients calculated for daily flow rates (mm d$^{-1}$) |
| RMSE | 2.04 | 1.72 | 0.24 | 2.92 | |
| Bias | 0.35 | 0.20 | 0.09 | 0.22 | |
| PBias | 35.11 | 19.63 | 9.33 | 21.73 | |
| RSR | 1.31 | 0.84 | 0.70 | 0.99 | |

b) coefficients for May to September

| Performance coefficients | Surface flow | Tile flow | WT (m) | Total flow | |
|---|---|---|---|---|---|
| NSE[*] | -18.98 | 0.19 | 0.40 | -11.76 | Coefficients calculated for hourly flow rates (mm h$^{-1}$) |
| RMSE[^] | 0.26 | 0.03 | 0.12 | 0.26 | |
| Bias[#] | -1.43 | 0.49 | 0.03 | 0.11 | |
| PBias[$] | -142.79 | 48.88 | 3.44 | 10.96 | |
| RSR[&] | 2.85 | 0.57 | 0.39 | 2.27 | |
| NSE | -3.89 | 0.21 | 0.41 | -1.08 | Coefficients calculated for daily flow rates (mm d$^{-1}$) |
| RMSE | 1.39 | 0.73 | 0.11 | 1.66 | |
| Bias | -1.43 | 0.49 | 0.02 | 0.11 | |
| PBias | -142.79 | 48.88 | 2.07 | 10.96 | |

| | | | | |
|---|---|---|---|---|
| RSR | 1.41 | 0.56 | 0.39 | 0.92 |



c) coefficients for October to April

| Performance coefficients | Surface flow | Tile flow | WT (m) | Total flow | |
|---|---|---|---|---|---|
| NSE* | -0.37 | 0.24 | 0.20 | -0.04 | *Coefficients calculated for hourly flow rates (mm h$^{-1}$)* |
| RMSE^ | 0.11 | 0.07 | 0.21 | 0.14 | |
| Bias# | 0.87 | 0.14 | 0.11 | 0.24 | |
| PBias$ | 86.59 | 13.56 | 11.00 | 24.11 | |
| RSR& | 0.90 | 0.67 | 0.77 | 0.79 | |
| NSE | -0.11 | 0.26 | 0.24 | 0.18 | *Coefficients calculated for daily flow rates (mm d$^{-1}$)* |
| RMSE | 1.50 | 1.56 | 0.21 | 2.40 | |
| Bias | 0.87 | 0.14 | 0.11 | 0.24 | |
| PBias | 86.59 | 13.56 | 10.58 | 24.11 | |
| RSR | 0.81 | 0.67 | 0.75 | 0.70 | |





[*] Nash-Sutcliffe efficiency, [^] Root-Mean-Square Error, [#] Model Bias, [$] Percentage Bias, [&] RMSE-observation standard deviation ratio

*3.5 Presence of capillary fringe: effects and hypotheses*
Results show that the thickness and vertical positioning of the capillary fringe have a strong
impact on the amount of drainable soil water that can flow into tiles. To investigate this effect
further, the response of tile flow and soil moisture to changes in the capillary fringe was
examined. It should be noted that although this thickness may change slightly depending on the
soil type and water retention curves (Skaggs et al., 1978), the model assumed a constant value
given the field-scale nature of the simulations and myriad of processes contemplated. However,
despite the simplification, the vertical positioning of the capillary fringe was still calculated and
enabled a dynamic (time-dependent) calculation of the drainable soil water that was available for
tile drainage over time.

*Effect of capillary fringe on tile flow*
Figure 7a relates the simulated normalized total cumulative tile flow ($Q_{tR}$ , total tile flow divided
by the total tile flow when there is no influence of capillary fringe) to capillary fringe drainable
water ($\varphi_{cR} = \varphi_c/\varphi_s$ ) for two different $\varphi_s$ values (0.045 and 0.125). The values were
normalized (0 - 1 scale) for comparison purposes. As expected, the model indicates that tile flow
increases with drainable water, but the relationship is non-linear, likely because as tile carrying
capacity is exceeded more frequently, there is more opportunity for groundwater seepage and
evapotranspiration. The direct effect of $\varphi_s$ (comparing the solid and dashed lines) on tile flow is
small because the amount of water that can effectively drain to the tile is controlled by the
capillary fringe and the associated drainable soil water. Figure 7b looks at the impact of the
capillary fringe thickness on tile flow. Here, the values are also normalized. Results show that
$Q_{tR}$ decreases with increasing normalized thickness of the capillary fringe, $T_{CFR}$ ($\frac{T_{CF}}{D_t}$ , capillary
fringe thickness divided by tile depth), but only while the $T_{CFR}$ is less than 1 that is when the
capillary fringe position is above the tile but has not reached the soil surface. Beyond this point,
increments in the capillary fringe thickness have no impact on tile flow because *Condition 1* has
been reached (see Fig. 2), which essentially means that the capillary fringe has reached the soil
surface. The match between the curves for two different $\varphi_s$ values shows that the changes in $\varphi_s$
does not influence the effect of normalized capillary fringe thickness and drainable water on
normalized tile flow. In Appendix D the sensitivity of cumulative tile flow and mean soil water
table elevation to different parameters are shown along with general approaches for evaluation of
the model parameters for new sites, the site with no tile flow and water table observations.

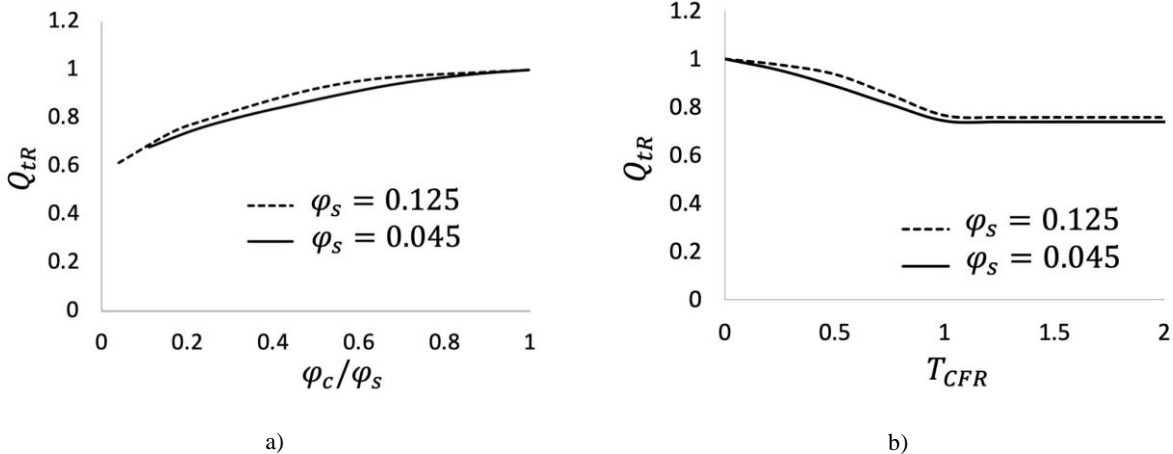

a)                                                          b)

Figure 7. Comparison between normalized tile flow ($Q_{tR}$) and (a) normalized drainable soil water ($\varphi_c/\varphi_s$) and capillary fringe

thickness ($T_{CFR}$) for different maximum soil saturation values ($\varphi_s$), by drawing the model prediction lines.


*Effect of capillary fringe on soil moisture*
Observations and model results of WT reveal a bimodal frequency distribution (Fig. 8 and 9,
respectively) with peaks at 0.85 m and 1.25 m depth, with the former corresponding to the depth
of the tile pipe and the second peak reflecting capillary fringe thickness. In the simulated soil
saturated storage (SSS as a measure of WT) frequency distributions (Fig. 9), the first peak
highlights again the efficiency of the tile in removing soil moisture. In contrast, the second peak
indicates a strong model response to differences in the capillary fringe thickness. It shows that
when there is near-constant percolation from the bottom of the soil layer, the matric potential
varies the greatest while it remains between the tile depth and the soil surface. While the water
table fluctuates faster and is more unstable within this range, it also remains there for shorter
periods. This bimodal response tends to push the water table depth below the tile. In Figure 9, we
can see that the first peak happens at 0.9 m depth where the tile pipe is located, and the second
peak happens at the depth equal to capillary fringe thickness. In Figure 9 the second peak is ore
clear for the capillary fringe thickness of more than 1000 mm. The first peak in the observed
water table frequency plot (Figure 8) happened around 0.8 m which almost matches with the tile
depth. And the second peak happened at the depth of ~1.2 m which shows that the capillary
fringe thickness should be around 1.2 m. But, to have a more reliable estimate for the capillary
fringe, based on Figure 8, data is needed at depths greater than 1.5 m.

The bimodal behaviour of the observed water table and simulated saturated storage demonstrated
here provides the opportunity to quantify the thickness of the capillary fringe using continuously
monitored water table elevations. The capillary fringe thickness determined using this method
can then be used as an input to the TDM module.

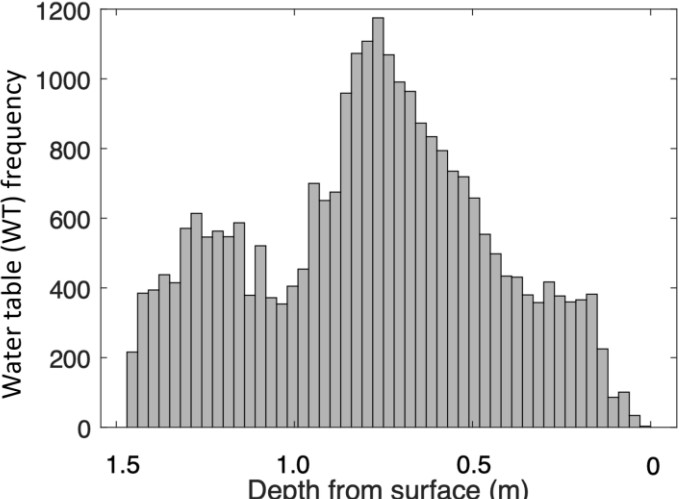


Figure 8. Histogram of the observed water table distribution for the period pf 2011 to 2018 in LON (Londesborough).


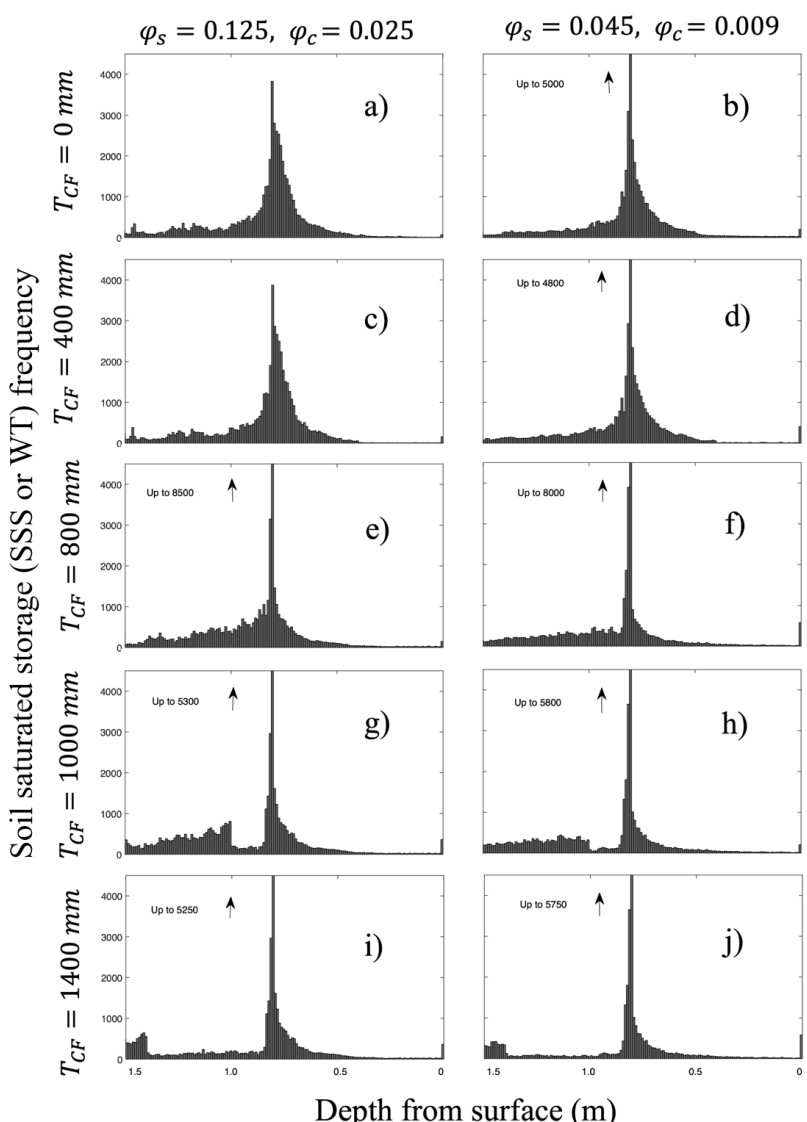


Figure 9. Histograms of the simulated soil saturated storages (SSS or WT) for the capillary fringe thicknesses of 0 (a,b), 400
(c,d), 800 (e,f), 1000 (g,h) and 1400 (i, j) mm and for the $\varphi_s$ and $\varphi_c$ of 0.125 and 0.025 (left column)as well as 0.045 and 0.009
(right column).

**4. Discussion**

*4.1    Insights into key control mechanisms of tile flow for catchment-scale simulations*
The model suggests that tile flow may not be accurately predicted exclusively based on the water
table depth and soil saturated hydraulic conductivity as suggested by the steady-state flow
assumptions of the Hooghoudt's equation (Hooghoudt, 1940). These results indicate two
additional controls: (1) the amount of drainable soil water in the soil, which has also been
identified in some field studies (*e.g.*, Skaggs et al., 1978; Moriasi et al., 2013) and (2)
fluctuations in the groundwater table (GWRD) are important to account for in catchment-scale
simulations. However, the relationship between drainable water and tile flow rates is non-linear,
as demonstrated in Fig. 7a. This is because the residence time for groundwater seepage and
evapotranspiration increases when the hydraulic tile carrying capacity is exceeded.
Comparatively, the effect of soil drainable water, $\varphi_s$ (see also Fig. 7a) on tile flow is small
because the capillary fringe and associated drainable soil water control the amount of water that
can effectively flow to the tile.

The verification of the model also indicated that the slopes of the rising and falling limbs of tile
flow hydrographs and WT were very sensitive to (1) the ratio between K and drainable soil
water; and (2) the net outflow in the soil through tile flow and groundwater level fluctuations
(GWRD).  This is supported by previous studies showing rapid responses of tile flow to
precipitation events (Gentry et al., 2007; Smith et al., 2015) and others that have related rapid
responses in tile discharge to antecedent moisture conditions (Macrae et al., 2007; Vidon and
Cuadra, 2010; Lam et al., 2016a; Macrae et al., 2019), which can be affected by the development
of a capillary fringe and its non-drainable water.

Results show that large fluctuations in WT (or SSS) and tile flow during the cold season, when
the water table tends to be above the tile, are primarily triggered by the development of a
capillary fringe that reduces the amount of drainable soil water. Model sensitivity tests showed
that a small amount of drainable soil water produces steeper rising and falling responses (and
with larger fluctuation amplitudes) in both the water table (saturated storage) and the tile flow.
Indeed, this pattern can be observed by exploring differences in tile drain responses in clay loam
soils with larger field capacities (and correspondingly smaller drainable water) and smaller
hydraulic conductivity which are more likely to experience pronounced oscillations (*e.g.*, steeper
rising and falling response curves) compared to tile drain responses of sandy soil, which is
characterized by reduced capillary forces, lower field capacities (but correspondingly larger
drainable water) and higher hydraulic conductivity. Notably, both model and observations of WT
(as a proxy for soil moisture) reveal a bimodal (*i.e.*, two peaks) frequency distribution when
examined in relation to the tile depth and capillary fringe thickness (Fig.  8 and 9, respectively).
The two peaks (*i.e.* most frequently observed WT or SSS conditions) correspond with the (1)
depth of the tile pipe (0.75 m), which demonstrates the efficacy of the tile at rapidly removing
excess soil water, and the (2) the capillary fringe thickness (for the depths of 1.0 and 1.4 m, Figs.
g, h, i and j) beyond which the amount of drainable water above the water table significantly
increases.

These findings align well with studies such as Lam et al. (2016a) that recorded soil moisture near
saturation after tile flow had ceased, suggesting the development of a capillary fringe. Combined
experimental and modeling works, such as in Moriasi et al. (2013) and Logsdon et al. (2010),
also discuss the impact of drainable soil water ("drainable porosity" or "specific water yield") on
tile flow and note that the drainable water is, in turn, dependent on the soil type, soil-water
dynamic and water table depth. However, these studies did not explore the dynamic nature of the
capillary fringe and its thickness relative to the soil column above in determining the transient
amount of drainage soil water that will impact the WT distribution and tile flow differently over
time (*Conditions 1* to *3*, see Fig. 2). Herein, while a capillary fringe with a fixed thickness that is
generally related to the soil properties was assumed, its vertical positioning was simulated
dynamically, which allowed determining the drainable soil water based on the evolution of
pressure head corresponding to field capacity.  Thus, the development of the TDM has provided
a step forward in the modeling of tile drainage and suggests that in loam soils such as those at the
study site, the effects of a capillary fringe on tile flow should be included. Soil moisture (soil
unsaturated storage) measurements from the study site by Van Esbroeck et al., (2017) between
November 2011 and May 2014 from depths of 10, 30, and 50 cm (using EC-5 Soil Moisture
Smart Sensor) showed that almost 90% of the gravitational soil moisture drains out with 0.5 to
2.5 h. It suggests that the water table and capillary fringe can reach an equilibrium condition
within one hour at this field site, enabling us to use a steady state equation (Hooghoudt, 1940) to
predict the dynamic behavior of the water table fluctuations.

*4.2    Importance of capturing seasonal patterns in groundwater to improve tile flow*
*predictions*
The GWRD changed dramatically between seasons affecting soil moisture (both saturated and
unsaturated storage of the soil) and tile flow patterns. Both observations and model results show
that low precipitation and higher evapotranspiration rates tend to produce little tile flow during
the growing season. These seasonal patterns in precipitation and evapotranspiration are
accompanied by a reduction in soil moisture (both unsaturated storage and saturated storage )
that leads to a substantial storage capacity in fields. Even following moderate and high-intensity
storms during the growing season, rapid soil moisture increases are observed (both saturated and
unsaturated soil storage); however, tile flow rarely develops, suggesting that the soil is able to
hold the water (Lam et al., 2016a; Van Esbroeck et al., 2016). In contrast, tile flow is often
observed during the cold season, even during smaller rainfall-runoff and snowmelt events
because of reduced soil storage but also a seasonal increase in GWRD (Lam et al., 2016a;
Macrae et al., 2007, 2019; Van Esbroeck et al., 2016). This concurs with several studies
throughout the Great Lakes and St. Lawrence region that have reported stronger tile responses
during the non-growing season, with the summer months often showing little to no tile flow
(Lam et al., 2016a, 2016b; Jamieson et al., 2003; Macrae et al., 2007; Hirt et al., 2011; King et
al., 2016; Van Esbroeck et al., 2016; Plach et al., 2019).

These results (the controlling effect of soil drainable water and groundwater level fluctuations on
tile flow) suggest that while soil moisture (both saturated and unsaturated storage) is largely
controlled by tile flow rather than GWRD in the cold season, this reverses in the growing season
(*i.e.,* soil moisture controls tile flow), with soil moisture (both saturated and unsaturated storage)
being also impacted by evapotranspiration. The controlling effect of groundwater fluctuations in
the growing season has also been studied by Hansen et al., (2019). The model indicated that the
rapid drops in observed WT during the growing season could not be explained by
evapotranspiration alone as well as the crop root depths, thus pointing to the role of GWRD.
Johnsen et al. (1995) and Akis (2016) also showed that the effect of groundwater accretion was
more effective on tile flows than surface runoff. Also, Vaughan et al. (1999) found that tile drain
flows in their study site in San Joaquin Valley of California were better explained and related to
nonlocal groundwater appearance than to local variations in irrigation amount,
evapotranspiration, variation in water storage or tile drain blockage. Thus, it was determined that
in addition to soil saturated hydraulic conductivity and soil thickness, the seasonal groundwater
fluctuations and capillary fringe drainable water are other important controlling factors on tile
flow rates.

## 711    5.      Conclusion

A new tile drain module within the modular Cold Regions Hydrological Modelling (CRHM)
platform has been created and tested at the field scale to support the management of agricultural
basins with seasonal snow covers. The model was tested and validated for a small working farm
in southern Ontario, Canada, and presents a step forward in the dynamic simulation of tile flow
and its effects on the hydrological cycle in cold climates. Observations and model results showed
that the dynamic prediction of tile flow and soil moisture at catchment scales needs to account
for (1) the amount of drainable soil water that can be affected by the development of a capillary
fringe and (2) fluctuations in the groundwater water table, in addition to the typical (3) water
table elevation above the tile pipe and (4) the soil saturated hydraulic conductivity considered by
the steady-state flow Hooghoudt's equation.
The groundwater table and matric potential changed dramatically between seasons, affecting
patterns of soil moisture and tile flow. Observations and model results showed that low
precipitation and higher evapotranspiration rates caused minimal tile flows during the crop-
growing season. Conversely, tile flow was often observed during the cold season, even during
small rainfall-runoff and snowmelt events, due to a seasonal increase in the groundwater table
and soil-saturated storage.
Model sensitivity tests showed that the capillary fringe strongly affected the amount of drainable
soil water flowing into the tile. Tile flow increased with drainable water, but the relationship is
highly non-linear likely because, as the tile carrying capacity is exceeded more frequently, there
is more opportunity time for groundwater seepage and evapotranspiration. Finally, observations
and model results reveal a bimodal soil-saturated storage response in the presence of tiles, which
is controlled by the relative positioning of the capillary fringe in relation to the soil surface and
the depth of tile drains below the soil surface. Capturing these dynamics is a critical advance
enabling the accurate prediction of the swift hydrological changes caused by the presence of tiles
in models.
The TDM was developed as a first approximation from a single field site. Given this limitation, it
is not yet widely applicable across multiple field sites. However, the development of this module
provides critical insights into its potential and performance for hourly time-step simulations, as
well as the importance of regional groundwater table fluctuations and simplifying the capillary
fringe parameters within models in some landscape types. Future work will include building on
the model and adapting it for different soil textures, such as those in clay loam soils, where
preferential flow can have a strong impact on soil-saturated storage and tile flow. Also, explicit
representation of unsaturated flow will be needed to enable the use of the model regions where
groundwater is disconnected from surface water, as commonly happens in arid and semi-arid
regions. Subsequent steps include also the integration of the new TDM model with CRHM's
water quality modules.

## Code/Data availability

The tile flow and soil water table data are not publicly available and will be provided upon
request to the data owner, Merrin Macrae. TDM code is not completely implemented in the main
version of the Cold Regions Hydrological Model platform and is provided only upon request to
the corresponding author.

## Author contribution

MK and DC developed the model code and performed the simulations. MM prepared the data
and supported the field work. MK, DC and MM prepared the manuscript with contributions from
JP and RP. All authors edited the manuscript.

## Competing interests

The contact author has declared that none of the authors has any competing interests.

## Acknowledgements

Funding for this project was provided by the Canada First Excellence Research Fund's Global
Water Futures programme through its Agricultural Water Futures project. Funding for the
collection of the field data was provided by the Ontario Ministry of Agriculture, Food and Rural
Affairs. The support of the Biogeochemistry Lab at the University of Waterloo for the collection
of field data and of Tom Brown and Xing Fang of the Centre for Hydrology at the University of
Saskatchewan for CRHM development and updates is gratefully acknowledged. The Maitland

Valley Conservation Authority is thanked for providing some precipitation, rainfall, and
temperature data.

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

## Appendix A

Table A1. Instrument names and descriptions

| Instrument name | Description |
|---|---|
| Hach Flo-tote and FL900 logger | Flow velocity and water level measurement |
| U20, Onset Ltd. | Barometrically-corrected pressure transducer |
| Temperature Smart Sensor S-THB-M002 | Air temperature measurement |
| Wind Smart Sensor S-WSET-M002 | Wind speed measurement |
| (Silicon Pyranometer)-S-LIB-M003 | Solar radiation sensor |
| Tipping bucketrain gauge, 0.2 mm Rainfall Smart Sensor – SRGB-M002 | Rainfall measurement |
| RH Smart Sensor(S-THB-M002) | Relative Humidity measurement |




## Appendix B

Table B1. Parameter names and their symbols in CRHM platform

| Parameter symbol | Parameter name |
|---|---|
| Tair | Air temperature |

| Wspeed | Wind speed |
|---|---|
| RH | Relative Humidity |
| Qsi | Incoming solar irradiance |
| R | Rainfall |
| WQ_soil | Water Quality soil module |
| WT | Water table elevation above the semipermeable layer |
| SSS | Soil saturated storage or the saturated part of the soil moisture |
| soil_moist | Soil moisture |
| Poro_soil | Soil porosity |
| AL | Above layer |
| BL | Below layer |
| GWRD | Groundwater level fluctuations, groundwater recharge and discharge |




**Appendix C**

Here, it was shown how seasonal factors ($f_{y,i}$) is assessed for different years. Equation (4) can be
written as:

$G_{y,i} = G \times f_{y,i}$ $\hspace{6cm}$ (C1)

For each year ($y$), $f_{y,i}$ for the first ($f_{y,1}$) and second ($f_{y,2}$) part of the sine function ($G$) were
assessed individually. It should be note that in first and second part of the sine function for each
year  G is larger than zero ($G \geq 0$) and smaller than zero ($G < 0$),  respectively. $G$ can be defined
for the two parts as:

$\begin{cases} if\ G \geq 0\ \ [i = 1\ ]then\ f_{y,1} = x \\ if\ G < 0\ \ [i = 2\ ]then\ f_{y,2} = y \end{cases}$          (C-2)

$G$ is the sine function representing the annual fluctuations in water table (WT/SSS) or it can be
simply defined as the percolation rate (in  mm hr$^{-1}$) of soil water to groundwater through lower
semi-permeable layer. So, for $n$ years there are $n \times 2$ $f_{y,i}$ values. The default values for $f_{y,i}$ are 1
and the default values can be changed for each year and for first and second parts in each year
independently. Calculated $G_{y,i}$ in each time step add or subtracted to or from the total soil
moisture depend on its sign. The $f_{y,i}$ values for the sine function parameters are presented in Fig.
C1. The verified sine function time series along with time series of temperature, precipitation and
calculated evapotranspiration are shown in Fig. C1. In this figure it is obvious that in years 2012
and 2015 to 2017 the warm season amplitudes are larger. The ET values are happened more in
the warm seasons (growing seasons). Also, the seasonal oscillation in sine function is very
similar to the temperature general oscillations.

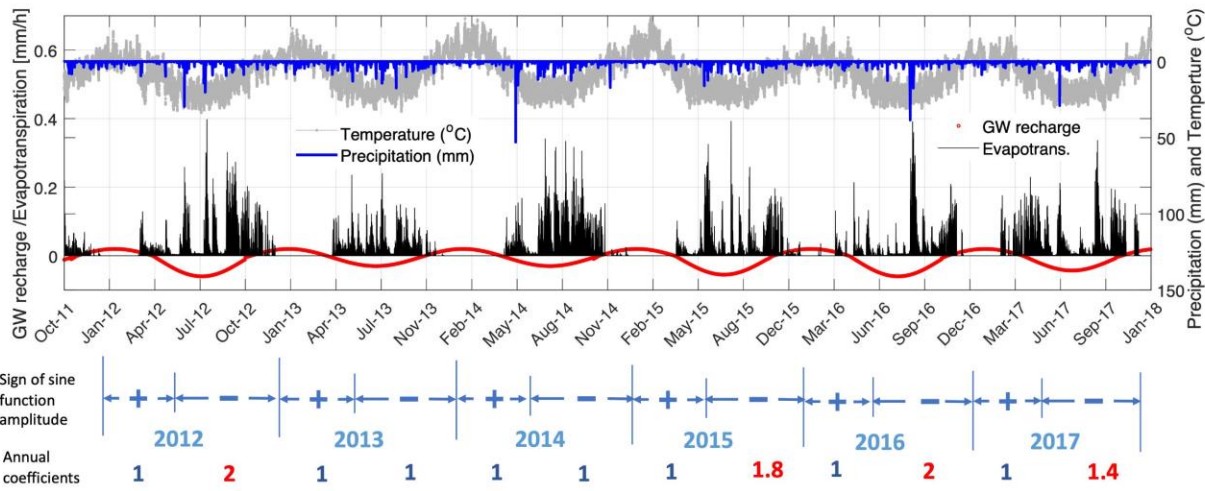


Figure C1. Time series of the adjustable sine function along with the time serioes of calculated evapotranspiration, temperature
and precipitation during the study period from Oct 2011 to Sept 2018.


**Appendix D**
A sensitivity analysis was conducted for the cumulative tile flow ($Q_{tc}$), mean soil water table
elevation ($WT_m$) and cumulative outflow rate from the semi-permeable layer at the bottom of the
soil to groundwater ($G_c$) (see section 2.4.5, Eq. 4) with respect to six module parameters.
Additionally, an approach for assessing model parameters at a new sites, potentially lacking
water table elevation and tile flow observations is proposed.

**D.1 Sensitivity analysis**
In this section, the sensitivity of $Q_{tc}$, $WT_m$ and $G_c$ to six distinct module parameters, namely
capillary fringe thickness ($T_{CF}$), capillary fringe drainable water ($\varphi_c$), soil saturated hydraulic
conductivity ($K$), soil thickness ($T_{SL}$), sine function amplitude ($A$) and sine function ($B$) was
examined. $Q_{tc}$, $G_c$ and $WT_m$ were computed over the entire simulation period, expressed in units
of mm, mm and m, respectively. Figures D-1a to f illustrate these sensitivities, with each
parameter's impact discussed in dedicated sections.

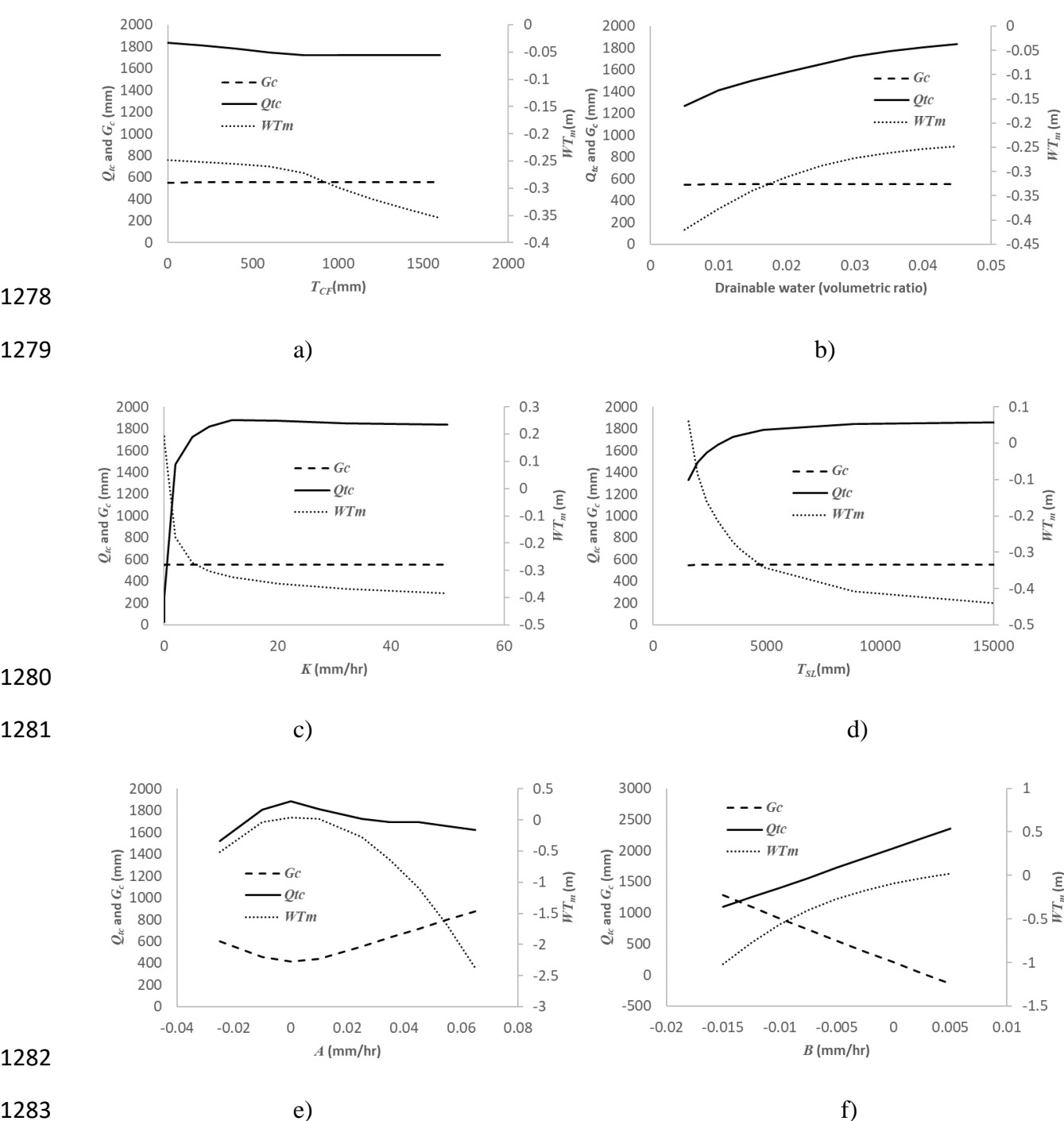


a)                                    b)

c)                                    d)



e)                                    f)

Figure D-1 Sensitivity of cumulative tile flow, $Q_{tc}$, cumulative soil to groundwater percolation rate, $G_c$, and mean soil water table
elevation, $WT_m$, to capillary fringe thickness, $T_{CF}$ (a) capillary fringe drainable water, $\varphi_c$ (b), soil hydraulic conductivity, $K$ (c),

soil thickness, $T_{SL}$ (d), sine function amplitude, $A$ (e ) and sine function intercept, $B$ (f).


*D.1.1 Sensitivity to capillary fringe thickness*
To gauge sensitivity to capillary fringe thickness $T_{CF}$, flow rates and the $WT_m$ were analyzed for
$T_{CF}$ ranging 0 to 1600 mm. Figure D-1a indicates that as $T_{CF}$ increases, both cumulative tile flow
($Q_{tc}$) and mean soil water table ($WT_m$) decline. The $WT_m$ drop is sharper for $T_{CF}$ beyond 900
mm. Beyond this thickness, $Q_{tc}$ stabilizes at a minimal value. A negative $WT_m$ indicates its
position below the tile pipe. $G_c$ remains consistent despite $T_{CF}$ variations.

*D.1.2 Sensitivity to capillary fringe drainable water*
With rising $\varphi_c$ both $Q_{tc}$ and $WT_m$ surge (Figure D-1b). As $\varphi_c$ ascends from 0.005 to 0.45, $Q_{tc}$
jumps from 1300 mm to 1900 mm and $WT_m$ from -0.45 m to -0.25 m (Figure D-1b). $G_c$ stays
constant, irrespective of $\varphi_c$ fluctuations.

*D.1.3 Sensitivity to soil hydraulic conductivity*
Increasing soil hydraulic conductivity ($K$) from 0 to 10 mm hr$^{-1}$ leads to a surge in $Q_{tc}$ and a drop
in $WT_m$ (Figure D-1c). However, adjusting $K$ from 10 to 50 mm hr$^{-1}$ results in leveling off slopes
for $Q_{tc}$ and $WT_m$, especially when $K > 20$mm hr$^{-1}$. Both metrics are acutely responsive to $K$ when
$K$ is below 10 mm hr$^{-1}$ but become non-responsive beyond 20mm hr$^{-1}$. $G_c$'s response to $K$
remains neutral.

*D.1.4 Sensitivity to soil thickness*
Similar to $K$, a rise in $T_{SL}$ from 1500mm to 15000 mm casue $Q_{tc}$ to rise and $WT_m$ to decline
(Figure D-1d). The most significant rate of change for both metrics occurs between 1500 to 5000
mm $T_{SL}$. Beyond 5000 mm, changes flatten. $G_c$ shows no response to $T_{SL}$ variations.

*D.1.5 Sensitivity to sine function amplitude*
Increasing the sine function amplitude, $A$, from -0.03 to 0 mm hr$^{-1}$ pushes both $Q_{tc}$ and $WT_m$
increase and reach to their maximum at $A$=0 (Figure D-1e). But as $A$ rises from 0 to 0.06 mm hr$^{-}$
$^1$, they both decline. In contrast, $G_c$ descends to its lowest (400 mm) when $A$ shifts from -0.03 to
0 and then increases to 900 mm as $A$ hits 0.063.

*D.1.6 Sensitivity to sine function intercept*
Both $Q_{tc}$ and $WT_m$ ascend with the growth in sine function's intercept, $B$. Increasing $B$ from -
0.015 to 0.005 mm hr$^{-1}$ sees $G_c$ descend. During this $B$ increase, $Q_{tc}$ expands from 1100 to 2400
mm, while $G_c$ shrinks from 1400 to 0 mm. It seems the sum of $Q_{tc}$ and $G_c$ might be constant.
This suggests that water either drains through the tile pipe or percolates through the soil bottom.
$Q_{tc}$, and $WT_m$ appear sensitive to all six module parameters, but $G_c$ only to $A$ and $B$.

**D.2 Module parameter evaluation for new sites**
As discussed in section 2.5, initial values for $K$, $T_{CF}$ and $\varphi_c$ can be determined by soil grain-size
distribution. Parameters less explored in past research for new sites include the sine function's
amplitude ($A$), intercept ($B$), and time delay ($D_d$).

*D.2.1 Evaluating sine function's A and B*
If no percolation exists from the soil's bottom to groundwater and $G_{y,i}$ is zero, both $A$ and $B$
should be zero. However, if percolation or interactions between soil and groundwater occurs, $A$
and $B$ need calibration assessment. Before this, reasonable initial values and bounds must be set.
From this study's findings, $A$ and $B$ should fall between the mean hourly difference of
infiltration and observed tile flow rates. For instance, observed hourly rates for infiltration and
tile flow at our site are 0.07 and 0.03 mm hr$^{-1}$. Thus, $A$'s and $B$'s initial values should range from
-0.04 to 0.04 mm hr$^{-1}$. Negative $A$ and $B$ values indicate outflow from soil to groundwater and
vice versa. Initial values were set at 10% of the range limits: -0.004 for $B$ and 0.004 for $A$.
Eventually, $B$ and $A$ were adjusted to -0.005 and 0.025 mm hr$^{-1}$.

*D.2.2 Assessment of sine function's time delay*
The sine function begins on the first Julian day. If its peak occurs around 91$^{st}$ Julian day ( three
months later), its minimum should be on the 274$^{th}$ day. If the peak comes later, say the 111$^{th}$ day,
a 20-day delay is present. This delay should mirror in both function's minima and maxima. In
this case the minimum would be on day 294. This delay aligns with the soil water table's peak
annual fluctuations. When no observed fluctuations exist, the delay can be calibrated. A sensible
initial delay can be ascertained by examining the study site's water table elevations, fitting a sine
function, and noting the peak's Julian day annually.