# Peer review of "Developing a tile drainage module for Cold Regions"

_EGUsphere, 2023_

## Referee Comment (RC1)

General comments

The manuscript entitled "Developing a tile drainage module for Cold Regions Hydrological Model: Lessons learned from a farm in Southern Ontario, Canada" submitted to EGUsphere introduces and evaluates a new tile drainage module for estimating surface and tile drainage flows using an 8-yr surface and tile drainage data set from a farm in southern Ontario. While the manuscript is generally well written, some important clarifications are needed throughout. Consistent terminology should be applied to soil water and groundwater flow processes. For example, the term 'soil water levels' is frequently used but needs to be consistently defined in the overall conceptual model of water flow. In several sections it is unclear whether the authors are implying unsaturated (e.g., matric potential) or saturated (shallow groundwater, Darcian flow) flow conditions. While considerable time is spent on the importance of the capillary fringe, there is an overall lack of discussion focused on the model predictions versus field flows and how they were influenced by annual weather variation or seasonally driven factors. It would also be beneficial for readers to see how the model performs for some signature (large) runoff events (maybe include some snowmelt events since the model accounts for these processes). It is not clear how modeled groundwater dynamics were used to adjust tile drain flows. The abstract does not mention how the module performed for the field data. It also mentions lateral flow to tiles but there is no relevant discussion of the results. Considerable time is spent on introducing the new module in the introduction compared to the relatively short time discussing model effectiveness for the field data. It would be beneficial to present some "next steps" for TDM module development based on the present findings and vision for further calibration/validation and improvement. Overall, the ideas and data presented by the authors are novel and well-conceived. The paper will benefit from a hearty revision. Please refer to line-by-line comments below.

Abstract
The tile drainage module's effectiveness for predicting field tile flows should be added. A stronger concluding sentence based on the implications of your most significant finding is suggested. Line 27: "Water level patterns". Clarify 'water levels' here and throughout, i.e., groundwater (saturated/gravity driven flow) or soil-water under tension (unsaturated, under tension/Richard's eq.)

Introduction

Line 116: Clarify "soil water level". Saturated or unsaturated conditions?

Lines 118-126.

Line 147: Does "pressure head" refer to groundwater elevation head?

Methods

Line 158-159. Tile water may include shallow groundwater flows not exclusively "infiltrated" from the study field.

Line 166-167. Clarify the last sentence more. How are surface and groundwater isolated for measuring and sampling if they outlet in the same area?

Lines 188-191: What's the size range of HRUs that can be modeled?

Line 207: "…soil water level (water table position). Please clarify. Is soil water level synonymous with water table elevation head?

Line 232: 31. 7 km seems far for assuming similar precipitation patterns.

Line 265: Clarify "soil water level observations"

Line 283: Clarify "water level" in conditions 3. Groundwater level?

Figure 2. The schematic looks good but it is slightly hard to follow.

Line 322: "…water level from soil moisture". Groundwater elevation head level?

Line 324: "…and was calculated as…"

Line 335: How is 'bottom of the soil' determined? Same as tile depth?

Line 362: $G_{y,I}$ is not defined in variable explanations. How were changes in groundwater elevation used to modify tile drain flows?

Line 387: Five different methods

Line 411: Suggest combining results and discussion

Figure 4. Add a period after "Figure 4". This graph looks good. Maybe enlarge somewhat if possible. SWE needs to be defined in the caption.

Figure 5. On the y-axis you have "Groundwater/soil water level (mm)", assuming they are indeed synonymous. In the caption, however, you refer to it as "water level in the soil". As stated, please revise this terminology consistently throughout to avoid the confusion.

Lines 456-457. Did you evaluate seasonal or interannual variability of model predictions? What about looking at other times when there was either good or not so good agreement between observed and modeled flows?

Table 2. Add a period after "Table 2". Add time period over which modeling represents. Assuming it included all events over the monitoring period? What about interannual variation? Add a footnote explaining model performance acronyms (NSE etc.).

Figure 7a,b: Are these best fit lines or model predictions? Please clarify.

Figure 8. Add period. Define LON.

Figure 9. Clarify "soil water levels" in caption. Matric potential?

Line 552. It is not clear if groundwater is being used to adjust tile flows. How are SWL, groundwater elevation head and matric potential related over a range of soil moisture contents?

Line 554. Suggest revising the term "opportunity time" to residence time or a similar term

Line 579. Clarify SWL (matric or elevation head?)

Line 592. "soil water level depth". Clarify

Line 599. Is one field representative of catchment scale hydrology?

Lines 624-629. More discussion should be added here groundwater dynamics and relationship to capillary fringe and tile flows. Including some recent references and how they relate to your findings would also benefit the paper.

Line 632. Modeling flows in one field does not capture the myriad of conditions found in larger, more complex catchments. Also, how does the model handle the high variability of saturated hydraulic conductivity, porosity and other spatially variable inputs? What about preferential flow to tile drains? How was this handled for the field site and was it a substantial component of overall flow (macro vs. micropore flow?).

Line 656. Do you mean the depth of tile drains below the soil surface?

---

## Author Response (AR1)

**RC1: 'Comment on egusphere-2023-142', Anonymous Referee #1, 11 Apr 2023**

https://doi.org/10.5194/egusphere-2023-142-RC1

**Referee #1, General comments:**

General comments

The manuscript entitled "Developing a tile drainage module for Cold Regions Hydrological Model: Lessons learned from a farm in Southern Ontario, Canada" submitted to EGUsphere introduces and evaluates a new tile drainage module for estimating surface and tile drainage flows using an 8-yr surface and tile drainage data set from a farm in southern Ontario. While the manuscript is generally well written, some important clarifications are needed throughout. Consistent terminology should be applied to soil water and groundwater flow processes. For example, the term 'soil water levels' is frequently used but needs to be consistently defined in the overall conceptual model of water flow. In several sections it is unclear whether the authors are implying unsaturated (e.g., matric potential) or saturated (shallow groundwater, Darcian flow) flow conditions. While considerable time is spent on the importance of the capillary fringe, there is an overall lack of discussion focused on the model predictions versus field flows and how they were influenced by annual weather variation or seasonally driven factors. It would also be beneficial for readers to see how the model performs for some signature (large) runoff events (maybe include some snowmelt events since the model accounts for these processes). It is not clear how modeled groundwater dynamics were used to adjust tile drain flows. The abstract does not mention how the module performed for the field data. It also mentions lateral flow to tiles but there is no relevant discussion of the results. Considerable time is spent on introducing the new module in the introduction compared to the relatively short time discussing model effectiveness for the field data. It would be beneficial to present some "next steps" for TDM module development based on the present findings and vision for further calibration/validation and improvement. Overall, the ideas and data presented by the authors are novel and well-conceived. The paper will benefit from a hearty revision. Please refer to line-by-line comments below.

***Response to general comments***:

We thank the referee for these insightful comments.

Water table terminologies were made consistent throughout the manuscript, and unsaturated and saturated flow conditions were clarified based on the provided detailed comments.

More discussion on the effectiveness of the model in the prediction of tile flow and how seasonality affects tile flows and creates gaps in the growing season was added to both the abstract and results/discussion. In addition, the good match between simulated and observed flows is shown in Figures 4 and 5.

In response to "How the modeled groundwater dynamics were used to adjust the tile drain flow", it should be mentioned that, as discussed in comment #16, groundwater is fully connected with soil water, such that the soil water table and groundwater table are synonymous in this study. It means that when groundwater fluctuates, we will have exactly the same fluctuations within the soil water table (WT or soil saturated storage, SSS). Fluctuations in WT affect the tile flow since tile flow here is calculated based on the Hooghoudt (1940) equation and using soil water table elevation. So, a change in groundwater table causes corresponding changes in soil water table and tile flow. The terminology "water table" is adopted throughout the manuscript now to refer to this combined saturated storage.

Finally, some next steps for improvement of the TDM module were added to the end of the conclusion section. Our responses to detailed comments are below.

**Referee #1 detailed comments** and our ***responses*** to comments are below (the line numbers in our responses are for the annotated or the track-changes version of the manuscript).

1) **Comment:**

The tile drainage module's effectiveness for predicting field tile flows should be added. A stronger concluding sentence based on the implications of your most significant finding is suggested. Line 27: "Water level patterns". Clarify 'water levels' here and throughout, i.e., groundwater (saturated/gravity-driven flow) or soil-water under tension (unsaturated, under tension/Richard's eq.)

***Response***:

Lines 23-26: A sentence about the effectiveness of the module in the prediction of field tile flow was added.

Lines 36-44: A sentence was added about the main findings of our research: "A novel aspect of this module is the use of …"

Lines 36-38: The sentence about "Soil water level (elevation head) histogram" was deleted.

**2) Comment:**

Line 116: Clarify "soil water level". Saturated or unsaturated conditions?

***Response***

Line 127-128: It was changed to "water table position".

**3) Comment:**

Lines 118-126.

***Response***

We assume that the reviewer is referring to the use of "water level depth" here. The term "water level depth" was changed to "depth of saturated water" (Line 131).

**4) Comment:**

Line 147: Does "pressure head" refer to groundwater elevation head?

***Response***

Line 169-170: "pressure head" was changed to "groundwater/soil water elevation head"

Methods

**5) Comment:**

Line 158-159. Tile water may include shallow groundwater flows not exclusively "infiltrated" from the study field.

***Response***

Lines 184-186: Changed to "The tile network collects infiltrated water from about 75% of the field (~ 7.6 ha), but may also receive lateral groundwater flow from neighboring fields."

**6) Comment:**

Line 166-167. Clarify the last sentence more. How are surface and groundwater isolated for measuring and sampling if they outlet in the same area?

***Response***

Lines 199-203: This has been clarified in this section. "However, surface water or elevated soil moisture conditions are not observed in this topographic low during smaller events or dry periods of the year, suggesting that this saturated ponding is not in a groundwater discharge zone. Although surface ponding is observed in the topographic depression within the field, water discharges freely at the opposite end of the culvert, facilitating the measurement of flow."

**7) Comment:**

Lines 188-191: What's the size range of HRUs that can be modeled?

***Response***

The size of TDM HRUs can be as small as the size of a single tile pipe (e.g., 1 m) times the pipe spacing (which was 14 m in our case study region), and as large as entire tile networks within a given farm or study area. We have added this sentence in L224 -226.

**8) Comment:**

Line 207: "…soil water level (water table position). Please clarify. Is soil water level synonymous with water table elevation head?

***Response***

Line 256: We changed the "…(water table position)…" to "…(water table elevation head)…"

**9) Comment:**

Line 232: 31.7 km seems far for assuming similar precipitation patterns.

*Response*

Lines 287-291: We have clarified the text in the paper, which was unclear in the previous submission. Rainfall was measured at the study site. Hourly snowfall was obtained from nearby stations as far as 31.7 km away.  These snowfall observations were tested for local application using periodic on-site snow surveys and found to be similar.  As snowfall varies far less over space than rainfall this provides reliable precipitation inputs to the model.

**10) Comment:**

Line 265: Clarify "soil water level observations"

*Response*

Line 327: We now consistently used "water table" which is the water level elevation we measured using an observation borehole.

**11) Comment:**

Line 283: Clarify "water level" in conditions 3. Groundwater level?

*Response*

Line 345: "water level" was changed to "water table"

**12) Comment:**

Figure 2. The schematic looks good, but it is slightly hard to follow.

***Response***

A legend was added to Figure 2, which makes it much clearer now.

**13) Comment:**

Line 322: "...water level from soil moisture". Groundwater elevation head level?

***Response***

Lines 383-385: The sentence was revised as: "To calculate a saturated storage (water table or groundwater elevation head level) from soil moisture calculated by the model, a threshold soil moisture content ($sm_t$) is defined, ..."

**14) Comment:**

Line 324: "...and was calculated as..."

***Response***

Line 387: The sentence has been revised.

**15) Comment:**

Line 335: How is 'bottom of the soil' determined? Same as tile depth?

***Response***

Line 399: The bottom of the soil was determined based on the soil cores and assessment of the depths of layers with significantly smaller hydraulic conductivities and finer grains compared to the shallow soil layer. In TDM, the soil bottom layer doesn't have to be completely impermeable; it can be set as semi-permeable with adjustable permeability.

**16) Comment:**

Line 362: $G_{y,l}$ is not defined in variable explanations. How were changes in groundwater elevation used to modify tile drain flows?

***Response***

Lines 424-425: A sentence was added to define $G_{y,i}$. The tile flow is calculated using the Hooghoudt equation (Hooghoudt, 1940) based on water table elevation in each time step (section 2.4.3). So, when seasonal groundwater/soil water table changes are added to the system, it will affect the tile flows accordingly.

**17) Comment:**

Line 387: Five different methods

***Response***

Line 452: This was corrected.

**18) Comment:**

Line 411: Suggest combining results and discussion

***Response***

The manuscript presents the results obtained for several hydrological variables, including tile flow, water table, surface flow, and total flow, as well as supplementary results and analyses, such as the results of a sensitivity study for the tile flow, soil water table dynamics, capillary fringe thickness and drainable water. For this reason, we decided to have the discussion section as an individual section as is normal practice in most published papers in hydrology. In the discussion section, we focused on discussing the results, describing the key controlling mechanisms of tile flows and the effect of groundwater seasonality.

**19) Comment:**

Figure 4. Add a period after "Figure 4". This graph looks good. Maybe enlarge somewhat if possible. SWE needs to be defined in the caption.

**Response**

A period was added after "Figure 4" (Line 488). SWE was defined in the caption. Figure 4b, with the shorter time interval, was added.

**20) Comment:**

Figure 5. On the y-axis, you have "Groundwater/soil water level (mm)", assuming they are indeed synonymous. In the caption, however, you refer to it as "water level in the soil". As stated, please revise this terminology consistently throughout to avoid confusion.

**Response**

We revised the caption to "… Saturated storage/ water table …". Also, Figure 5's left vertical axis title was revised.

**21) Comment:**

Lines 456-457. Did you evaluate the seasonal or interannual variability of model predictions? What about looking at other times when there was either good or not-so-good agreement between observed and modeled flows?

**Response**

As can be seen in Table 2, the performance of the model in predicting tile and surface flows, as well as soil water table, was evaluated for the entire simulation period. In Lines 533 to 536, we discussed the reasons for uncertainty in the prediction of cumulative surface flow and total flow (in Figure 6), which partly occurred due to the blockage of the tile pipe between 2017 and 2018. The main goal of the TDM module was to simulate tile flow and water table, and we can see in Table 2 that the model performance in the prediction of tile flow and the water table was significantly better than in the prediction of surface water flow. On the other hand, Figures 4 and 5 show an acceptable match between observed and simulated tile flow and water table, and we can see that TDM was successful in capturing large seasonal water table

fluctuations, their corresponding seasonal gaps in tile flow, and the sudden drops in water table.

We believed that by calculation and implementation of $G_{y,i}$ or annually variable seasonal groundwater fluctuation coefficient, we evaluated and considered the seasonal or interannual model prediction capabilities.

**22) Comment:**

Table 2. Add a period after "Table 2". Add time period over which modeling represents. Assuming it included all events over the monitoring period? What about interannual variation?

**Response**

A period was added after Table 2. The performance coefficients in Table 2 are calculated for the entire simulation period. Since the observed soil water table data did not cover the entire simulation period, we were not able to evaluate the interannual variations in the performance coefficients for soil water table. So, we decided to evaluate the coefficients for the entire simulation period only.

**23) Comment:**

Add a footnote explaining model performance acronyms (NSE etc.).

**Response**

Footnotes explaining 5 model performance coefficients were added to Table 2.

**24) Comment:**

Figure 7a,b: Are these best fit lines or model predictions? Please clarify.

**Response**

The lines in Figures 7a and b are the model predictions. We added to the figure caption that the lines are the model predictions.

**25) **Comment:**

Figure 8. Add period. Define LON.

***Response***

These changes have been made.

**26) **Comment:**

Figure 9. Clarify "soil water levels" in the caption. Matric potential?

***Response***

These are histograms of soil saturated storage (SSS) or soil water tables (SWT). We extract the soil water table from the time series and plotted them as a histogram.

**27) **Comment:**

Line 552. It is not clear if groundwater is being used to adjust tile flows. How are SWL, groundwater elevation head and matric potential related over a range of soil moisture contents?

***Response***

As mentioned earlier, the groundwater at the site and in neighbouring areas are directly connected to soil water in the site, both water sources contributing to saturated conditions in the water table. So, groundwater fluctuations in neighboring areas directly affect the water table in tile drained area and correspondingly affect the tile flow rate. That is because the tile flow is calculated based on Hooghoudt (1940) equation (Eq. 1), as it is discussed in Section 2.4.3. According to Hooghoudt equation, the tile flow rate is proportional to the difference between the water table elevation and tile pipe elevation. In TDM the regional groundwater fluctuations are adjustable

and defined as a sine function (Eq. 4). So, in TDM we can adjust the groundwater fluctuations amplitude in each separate year [using $G_{y,i}$ in Eq. 4] to have a better match between the observed and simulated tile flows and water table time series. This means TDM can adjust the seasonality in groundwater table in neighboring fields in order to have a better match between tile flows and soil water tables.

Additionally, eq. 3 in section 2.4.4. presents the relationship between soil moisture and water table in the soil.

**28) Comment:**

Line 554. Suggest revising the term "opportunity time" to residence time or a similar term

**Response**

Line 644: We changed it to "residence time".

**29) Comment:**

Line 579. Clarify SWL (matric or elevation head?)

**Response**

Line 670: It stood for soil water level (SWL), but, it was revised to water table (WT) or soil saturated storage (SSS) according to Eq. 3 and Table B1 (Appendix B).

**30) Comment:**

Line 592. "soil water level depth". Clarify

**Response**

Line 683: It is soil water table (SWT) depth, as it is clearly mentioned in the text (Eq. 3 and Table B1).

**31) Comment:**

Line 599. Is one field representative of catchment scale hydrology?

**_Response_**

Lines 690-691: What we meant here, is that since the CRHM platform can be used at catchment scale, the TDM within a large catchments can be considered as one or more HRUs and the interactions between TDM HRUs with other HRUs within the catchment can help us to have a better idea about the effect of tile drainage on other parts of the catchment.

Another important point is that by implementing the groundwater table seasonality within the module, we consider the effect of catchment scale groundwater fluctuations within the module. However, given the potential for confusion, we removed the words "at the catchment scale" from the sentence.

**32) Comment:**

Lines 624-629. More discussion should be added here on groundwater dynamics and relationship to capillary fringe and tile flows. Including some recent references and how they relate to your findings would also benefit the paper.

**_Response_**

Lines 717-730: More discussion has added about this topic with four new references added.

**33) Comment:**

Line 632. Modeling flows in one field does not capture the myriad of conditions found in larger, more complex catchments. Also, how does the model handle the high variability of saturated hydraulic conductivity, porosity and other spatially variable inputs? What about preferential flow to tile drains? How was this handled for the field site and was it a substantial component of overall flow (macro vs. micropore flow?).

***Response***

Line 733: To simulate large catchments with CRHM the catchment needs to be divided into a series of HRUs depending on the complexity of the catchment characteristics. We have revised the sentence to indicate that this work has been done for a single field site as a first step. Adding layer complexities and preferential flow paths in TDM are some of the next steps envisaged in the future as the TDM is adapted for other sites and soil textures such as clay soil, when preferential transport into tiles plays a more significant role.

**34) Comment:**

Line 656. Do you mean the depth of tile drains below the soil surface?

***Response***

Line 759: We revised the end of the sentence to "... and the depth of tile drains below the soil surface,"

RC2: 'Comment on egusphere-2023-142', Responses to Anonymous Referee #2, 1Jun2023

https://doi.org/10.5194/egusphere-2023-142-RC2

**Referee #2 general comments** and our *responses* to comments are below (the line numbers in our responses are for the annotated or the track-changes version of the manuscript).

**1) Comment:**

The topic is highly relevant and there is definitely a strong need to develop drainflow concepts and modules for catchment scale modelling. The proposed module seems reasonable builds on many sound observations regarding drainflow generation, although it depends on parameters that are typically not know in time or space outside the specific study site.

*Response*

In answer to the next comments we discussed how we can evaluate the required TDM module's parameters for areas out of the study site.

**2) Comment:**

I am quite sceptical about the observational data behind the model validation. Primarily because there is only one field site. Secondly, the water level and soil moisture is measured only in one location and on the edge of the field. Drainflow generation is highly variable from fields to field and within fields. I don't see how a single field drainflow record can validate a conceptual drainflow component of a model applicable to cold regions in general. In Denmark, we have recently collected a dataset of 45 timeseries of drainflow from tiledrained clay-dominated fields across the country. They display a large spread in drainfractions (relative to recharge or precipitation) and drainflow dynamics. I fear that the model developed here needs to be confronted with data from more than a single field to prove its general applicability.

*Response*

In this study we presented the preliminary version or the first test of TDM module and based on the available water table and tile flow data from a single site. The next phase of model development will test the module for more study sites and soil types as well as with more water table data from different locations within the site. We are aware of that the data from a single field cannot adequately validate a module, but we have learned a lot in this study and in a single site about how hourly simulation time steps, regional groundwater table fluctuation and simplifying the capillary fringe parameters

can work well in a real condition. However, as we mentioned above, we will test the module with more study sites and soil types in our future studies.

We have clarified this in the final paragraph of the conclusions section:
The TDM was developed as a first approximation from a single field site. Given this limitation, it is not yet widely applicable across multiple field sites. However, the development of this module has provided valuable insight into the potential for hourly simulation time steps, as well as the importance of regional water table fluctuations and simplifying the capillary fringe parameters within models in some landscape types. Future work includes building on the model and adapting it for different soil textures such as those in clay soils, where preferential flow can have a strong impact on soil saturated storage and tile flow. Also, explicit representation of unsaturated flow will be needed to enable the use of the model regions where groundwater is disconnected from surface water, as commonly happens in semi-arid agricultural regions.

**3) Comment:**

Also, during the argumentation for the TDM module assumptions, it is argued that ", which presumes no surface ponding, an assumption that generally holds at the study site (Eq. 1), where water ponds only during very wet periods and on a small portion of the study site" and again in line 307-308". But the TDM module most be more general than just fit for the specific study site. It is intended to be applied across Canada and cold regions beyond.

*Response*

The reviewer is correct. As noted in the previous comment, the TDM developed here was initially done for a single field site to improve our understanding of process and ability to test this knowledge with a robust data set. However, the next phase of model development will be to build on this early work to make it applicable across multiple landscapes. The reason for using only the equation for the condition with no surface ponding for the tile drainage was to keep the preliminary version of the model very simple.

**4) Comment:**

Also, several of the parameters required might be know at this particular site (e.g tile drain spacing, CF thickness, CF drainable water, seasonal factor, maximum soil

moisture and capillary fringe thickness) but how would they be parameterized in general when applied in a large-scale hydrological model?

*Response*

Some parameters such as drain spacing are usually publically available from governmental or non-governmental relevant organizations' websites. But for some others such as CF thickness, CF drainable water and maximum soil moisture, some initial values can be extracted using the soil texture as we discussed in our response to comment #12 from referee #2 (specific comments). We can use these extracted values for the parameters as initial values in the model. Then, these initial values can be adjusted during model calibration or verification by matching the observed and simulated tile flows and soil water table time series. The seasonal factors for each year also can be adjusted during model calibration/verification. Also, for the large-scale hydrological model the area can be divided to smaller HRUs and the parameters are defined and verified for each HRU separately.  Since most CRHM parameters describe physically based modules, they are physically identifiable and measurable. CRHM, without tiles, has been successfully applied to large agricultural areas in the Canadian Prairies with parameterization of each HRU from soil textural, land cover, drainage and topography maps and satellite data (Spence et al., 2012, HESS).

**5)  Comment:**

The list of parameters needed to parameterize the TDM shows how many parameters are needed and most of them needs to be calibrated/adjusted. It seems to be a very poorly constrained concept, given that a single drainflow time series is used to estimate all the parameters, and that such time series are typically not available outside dedicated study sites.  Table 1 should include some consideration on how parameters would be obtained outside the study site and in lack of observational data.

*Response*

In general, model parameter setting in CRHM is done without calibration, following the Deduction, Induction, Abduction approach outlined by Pomeroy et al. (2013).  As most parameters are physically identifiable and measurable, they can be determined from available geospatial datasets.  Some that are not can be abducted from research basins and sites such as this one to biogeophysically similar basins.  This is described in Pomeroy et al. (2022).  There are many ways to identify parameters in this model application. To estimate the TDM parameters other than with drainflow time series, we can also use either the water table time series or soil moisture time series. Soil

parameters can be estimated using the soil grainsize distribution or soil retention curve (if it is available). The seasonal factors for each year in this study area were evaluated by matching the general trends of the simulated tile flow rates and soil water table with the observed time series. To be able to adjust the seasonal factor in each year we need to have adequate amount of observation in each year, but these could be transferred to similar sites and catchments if needed.

In Lines 454-464 We have mentioned some sentences on parameter estimation techniques and how they were further adjusted to local conditions.

**6) Comment:**

Also, a sensitivity analysis of all parameters would be relevant to better understand what drives drainflow generation in the module.

*Response*

The key parameters that control the tile flow rate and water table in the soil and were the main focus of this study were the capillary fringe drainable water as well as the capillary fringe thickness. So, in section 3.5 (Figures 7 a and b) we analyzed the sensitivity of the total tile flow to these two key parameters. The sensitivity of the tile flow rate to other parameters such as soil hydraulic conductivity, tile spacing, tile depth have been analyzed in previous studies through development and application of Hooghoudt equation (Hooghoudt, 1940).

**7) Comment**

It is stated that " However, field soil water level observations show evidence of annual groundwater level periodicity/fluctuation (Rust et al., 2019) that are sinusoidal in nature and cannot be neglected"

But how will this observation and the described method for accounting for annual groundwater level fluctuations be transferable to other sites? Some other sites may have very limited connection to groundwater and some more than the study sites used here.

*Response*

The reviewer is correct. Indeed, the annual periodicity in groundwater tables is something specific for each site and should be evaluated for each individual study site.

In this initial iteration of the model, we demonstrated the importance of including this variable, as it can be important in some catchments. In areas that are not connected to groundwater, output from the soil is mostly through evapotranspiration or tile drainage, and the water table fluctuations are more controlled by the action of tile drainage. In such cases, we could adjust the seasonal factor for each year so that the general water table fluctuation trend would match observations.

As it is described in Appendix C, the annual periodicity in water table elevation in the groundwater layer is simulated by adding a sine function with annual wavelength to the simulated water table. The intercept and amplitude of sine function are separately adjustable for each year. In model verification step we adjusted the sine function parameter in order to have the better match between general trends in annual oscillation in simulated water table.

**8)  Comment:**

You seem to calibrate the groundwater contribution year by year, which would make it impossible to apply the approach outside fields where drainflow and water levels are already measured.

*Response*

As we discussed in Appendix C, to consider the water table fluctuations in the groundwater layer, we need to simply adjust the sin function parameters for each year. To adjust the sine function intercept and amplitude we need to compare the sine function to the general trend in observed soil water table time series. Or we can simply calibrate/verify the sine function parameters along with other parameters during model adjustment.

**9)  Comment:**

I am sorry to be critical here, but if you need drainflow and water level measurements for a specific field site in order to parameterize the model, how can it predict drainflow elsewhere?

*Response*

The objective of the modelling here is to better understand and diagnose the sub-surface hydrological system in tile-drained fields and catchments. This does not preclude applications elsewhere, as have been demonstrated with CRHM. Since TDM is a physically based module and works based on soil physical and hydraulic concepts and laws, for areas with no observed tile flow and soil water table we can evaluate the module parameters based on the soil characteristics which can be assessable based on soil grain size distribution or soil water retention characteristic curve. Based on the evaluated soil parameters and other input hydrological component, the module can assess the tile flow rate and soil water table time series. However, we strongly suggest to compare the model results with the observed tile flow or soil water table time series or sparse data.

**10) Comment:**

Given the ambition of incorporating this model in the CRHM for large scale applications, I feel that the model development is based too much on a single study site and cannot be validated sufficiently with data from just this one field. In addition, there is not enough consideration on the number of parameters required and how these would be estimated outside the study site.

*Response*

The referee has made a good point. The module development was based on a single study site and that was because it was our first step through developing a tile drainage module with time steps smaller than daily, with assessment of soil water table position (soil saturated storage) based on the calculated soil storage and by considering the regional groundwater table fluctuations. We faced many challenges in implementing all these new assumption and simplification in the module and at this stage we needed just a preliminary step, to prove that this approach works for a real field site. Definitely, as we previously mentioned, we will improve the module and make it applicable for broader range of soil types and field site hydrological conditions, in our future works. This has been noted in the revised paper.

**Referee #2 specific comments** and our responses to comments are below.

**1) Comment:**

L17: rephrase "enables creating"

*Response*

Lines 17-19: It has been rephrased.

**2) Comment:**

L22: It is claimed that the evaluation is "A robust multi-variable, multi-criteria model performance evaluation strategy" but it is based on XXX a single …

*Response*

Lines 23-26: We deleted "robust" from the sentence to show that the approach was just a preliminary step or just a start to develop a multi-variable and multi-criteria approach in simulation of tile flow.

**3) Comment**

L25: What does "moisture interactions" mean?

*Response*

Line 33: By using "moisture interactions" we meant the sending and receiving of the moisture between root zone and groundwater.

**4) Comment:**

L110-111: "Thus, to provide reliable estimations of water from farmland via surface runoff and tile 111 flow, models must be able to predict soil moisture and the soil water level accurately"

*Response*

Lines 126-128: the sentence was rephrased to "Thus, to provide reliable estimations of water loss from farmland via surface runoff and tile flow, models must be able to predict soil moisture and the soil water table position accurately ( …"

**5) Comment:**

L125: consider rephrasing "appreciably"

*Response*

Lines 141-144: the sentence has been changed to "It has also been shown that despite the presence of tile drains, the soil above the tile may not drain all gravitational water following an event and may remain at or above field capacity ( ...".

**6) Comment:**

L112-139: I cant help noticing that most of the literature is from the 1970'ies. Is this really the most recent developments within his area?

*Response*

This section of reviewing the literature review is focused on fundamental equations and theories about tile drainage and tile flow that were developed during this period and not significantly changed or improved in recent years to our knowledge. More recent studies are mostly utilizing numerical models based on these fundamental theories.

**7) Comment:**

L166: How do you know that this is not as groundwater discharge zone?

*Response*

Lines 197-201: The sentence has been revised to "Water tends to accumulate in a topographic low in the field, in front of the field outlet during snowmelt or high-intensity rainfall events, presumably due to either surface runoff or return flow (see ponded area, Fig. 1b). However, surface water or elevated soil moisture conditions are not observed in this topographic low during smaller events or dry periods of the year, suggesting that this saturated ponding is not in a groundwater discharge zone."

**8) Comment:**

L199-201: Tile drains are typically used in clay dominated soils, why is it meaningful to develop a drainage model only for coarse textured soils?

*Response*

Although it is indeed true that tile drains are typically used in clay-dominated soils, they are also found in fields with imperfect drainage, such as many landscapes found in the Great Lakes and St. Lawrence regions (e.g. Michigan and Vermont, USA and Ontario, Canada). Less is known regarding tile drain hydrology in these landscapes relative to those found in clay soils. For this reason, as well as the availability of a detailed data set, this landscape type was chosen for our pilot study. Next steps for this work will include testing and adapting the model for clay-dominated landscapes.

This has been added to the Introductory section:

Lines 161-167: "In this study, a new tile drainage module was developed and incorporated within the physically based, modular Cold Regions Hydrological Modelling (CRHM) platform (Pomeroy et al., 2022) to enable hydrological simulations in tile-drained farm fields in cold agricultural regions. As a first iteration, the new module was developed for a field with sloping ground and loam soil with imperfect drainage. Such landscapes are common in the Great Lakes Region (e.g. Michigan and Vermont, USA and Ontario, Canada) and tile drainage in such landscapes has not been as widely studied as it has been in clay-dominated soil."

**9) Comment:**

L212: How is drain flow measured in a pipe if the outlet zone is typically flooded (figure 1)?

*Response*

Our revised manuscript provides more clarity on how things were measured and why (Lines 256-275). Briefly, although water ponded within the field, it generally flowed freely through the surface culvert, facilitating the measurement of flow. However, both depth and velocity were measured within the culvert to permit robust measures of flow. This has been noted in several locations throughout the manuscript.

**10) Comment:**

L253: delete "specific"

*Response*

Line 312: it was deleted.

**11) Comment:**

L253-254: How would these site-specific data ("information about the tile network, such as the tile depth, diameter and  spacing") ever be obtained outside the particular site you investigate here?

*Response*

Information regarding site-specific details regarding tile depth, diameter and spacing may be obtained directly from landowners or can be estimated based on standard design and installation guidelines for the region. This has been added to the paper. Moreover, in many regions, such site-specific information about the tile drainage network is freely available on the relevant organizations' websites (example, LIO in Ontario: https://geohub.lio.gov.on.ca/documents/cf961d62ee1345c7b191808c9d60a4d7/about ).

**12) Comment:**

L255: Are "CF thickness and CF drainable water" model parameters? And how are they obtained?

*Response*

This is described in section 2.5. We have now noted this in the paper in response to the reviewer comment (lines 317-318). We evaluated an initial value for CF thickness and CF drainable water using soil texture based on the relationships presented by Twarskawi et al. (2009). These authors introduced CF thickness as "pressure head at field capacity",

and CF drainable water can be assessed based on "saturation at field capacity". These initial values for CF thickness and CF drainable water were estimated using the soil texture and the relationships presented in Twarskawi et al., (2009). Then, through verification of the model and matching the model output and the observed soil water table and tile flow, the CF thickness and CF drainable water parameters were adjusted. This process is described in Section 2.5.

**13) Comment:**

Line 465: perhaps also indicate statistics based on daily values, this might be more comparable to most other studies.

*Response*

According to the reviewer's comment we calculated the performance coefficients based on the daily time intervals and added them in Table 2. However, the daily performance coefficients except for NSEs for surface flow and total flow did not change substantially. The higher values of daily RMSE (compared to hourly ones) is related to the units.

**14) Comment:**

L606 sec. 4.2: Given this importance, how would this seasonal pattern of groundwater be estimated outside the study site? Would this not require a "real" transient 3D groundwater flow model?

*Response*

As we presented in section 2.4.5, because based on the given references, we knew that the water table annual fluctuations in the groundwater layer usually follow a sine function pattern, we simplified the approach and simulated this seasonality using a sine function. The sine function's amplitude and intercept parameters are adjustable and can be evaluated during model verification/calibration. Verification/calibration can be done by matching the observed and simulated water table (including that in the soil layer) and tile flow rate. It is possible that this can be improved upon in future using a 3D transient groundwater flow model; however, the simple sine function used here provided a good first approximation for the development of this module. This approach is more efficient in dealing with a small study site, where we do not have sufficient information regarding groundwater in the neighboring areas.

However, for regional scale modeling, it would be more reasonable to have an evaluation about the groundwater annual fluctuation by running a 3D groundwater model in advance and use their outputs within TDM module.

**15) Comment:**

L632: I would not say that it is tested for catchment-scale simulations. It has only been tested for a single field scale site.

*Response*

Line 710: "catchment-scale " was changed to "field scale site". The respected reviewer is right, however the approach and methods we used here can be used for wide range of similar study site with almost the same conditions but definitely this version of TDM does not cover all types of sites with different soil types and hydrological conditions.

**Citation**: https://doi.org/10.5194/egusphere-2023-142-RC2

---

## Referee Report (RR1)

General comments

The authors effectively addressed many of the reviewer comments in their revision to improve the manuscript's clarity, however considerable revision is still needed before publication. Please refer to line-by-line comments that follow.

Abstract

Line 12. "…extensively in [poorly drained] agricultural lands…" Suggest adding 'poorly drained'.

Line 16. Clarify 'runoff'. Surface, subsurface, both?

Line 26. "Shorted" should be 'shorter'

Line 25-29. Consider condensing this sentence, a bit hard to follow

Introduction

The introduction is still long and could be more concise.

Lines 45-50. Lowering the seasonally high water table in poorly drained fields is the main function of tiles drains

Line 61. "…that can represent tile drainage". Consider being more descriptive re: tile drainage…what type of flows, matrix? Gravitational? What about surface runoff?

Line 76. "Since the use of tile drainage has become popular… ". Hasn't tile drainage been used extensively for decades? Do you have any recent tiel adoption trend data specific to Canada that you can cite?

Line 100. You mention catchment scale but your study is at the field scale, please clarify whether the aim of CHRM-TD is catchment or field scale.

Lines 104-108. What about the hydraulic gradient?

Line 110. Suggest changing "Many" to 'Some studies'. Also some of the references cited are quite outdated.

Line 140. Integrate the last sentence with the previous one.

Line 154-155. "…which are increasingly being artificially drained". Citation?

Line 162. Soil type or series, not texture.

Line 192. If snowmelt processes were accounted for they should be better explained. How was melting estimated? What about infiltration with partially frozen soils?

Line 197. "Water quality" needs further clarification. What specific nutrients? Sediment?

Line 223. Preferential flow is likely an important mechanism at your site since the texture is a clay loam. Preferential flow even occurs readily in silt loams. Maybe just state that you did not model it for this study and will be assessed in future studies?

Line 227. Our research shows that soil freezing still happens with snow cover, with depth and extent depending on snowpack depth and other radiative factors.

Line 241. How good was the regression relationship for the rating curve?

Line 245. You mention "forcing" with other covariates but do not present or discuss forcing data. I suggest not using this term unless you did use it to force the model.

Line 344. Was Ks measured or assumed?

Line 347. "…was used to estimate"

Line 376. "…into the sire from adjacent farms". Replace 'farms' with 'fields'

Line 386. Clarify "This approach" at the start of the sentence.

Line 389-392. This sentence is long and hard to follow, suggest revising.

Line 421. State somewhere that these methods were used to assess model accuracy

Table 1. Remove "Source" as a column heading if it is not used.

Line 452. Suggest revising to "…the near absence of flow"

Line 462-463. Suggest revising to" Although peak tile drainage flow was not always…"

Line 474. Saturated soil storage and water table depth appear to be used interchangeably, which causes some confusion. Suggest sticking with one term or the other if you are implying the same physical state or clarifying the use of both terms.

Figure 5. Same comment as above. Y-axis lists 'SSS' and 'WT'- suggest sticking with one or the other as per above comment.

Lines 506-509.  Provide more specifics about the "systematic issues" and provide some ideas on why the surface flow is not predicted well and how you plant to improve it.

Lines 513-526. Suggest providing additional context here. What are these collective differences suggesting about the overall model?

Figure 6. Might be helpful to present the R2 values for relationships.

Line 545. Change "have" to 'had' a strong influence"

Line 546. "…that flowed into tiles.

Table 2. Revise for consistent significant digits across the table

Line 591. "ore" should be 'more'

Line 598. Again, suggest sticking with either water table (WT) or saturated soil storage (SSS) but not using them interchangeably to avoid unnecessary confusion.

Figure 9 caption. Same comment as above- use either SSS or WT but not both. Also, should it be water table depth or just water table?

Line 612-624. Can you use the relationship you found between capillary fringe and drainable water to improve observed vs. predicted flows? While this might be too much to add to your results, discussing how one would use these data to improve CRHM-TD module seems like an important area to discuss.

Line 627. Should 'K' be "Ks" for saturated hydraulic conductivity?

Line 635. Same comment re: WT or SSS- use one or the other or explain reasoning for using both terms.

Line 648. Same as above.

Line 666-667. Similar comment as above- can you use the new relationships to improve flow predictions? How would this process unfold if you are not able to apply it now?

Line 669-670. What about the fact that tiles are 1 m deep and soil moisture was measured at 0.5 m?

Lines 681-692. Shouldn't the role of evapotranspiration be included here?

Line 721. Delete space between sentences.

Line 742. As previously mentioned, preferential flow is likely an important mechanism in your field given the texture is clay loam.

---

## Author Response (AR2)

Below are our responses to the comments from two referees regarding egusphere-2023-142, entitled: Developing a tile drainage module for Cold Regions Hydrological Model: Lessons from a farm in Southern Ontario, Canada, authors: Mazda Kompanizare et al.

The line numbers on our responses to the referees' comments are based on the latest annotated version.

**Referee #1**

**Suggestions for revision or reasons for rejection**

Please refer to line-by-line comments. Line numbers refer to the track changes manuscript version,

**1)Comment**

Line 232: TDM defined already? Not until line 299

***Response***

Thank you for catching this. We added a definition for TDM in introduction section (Line 142-145).

**2)Comment**

Line 248: the soil is a clay loam

***Response***

In Line 223, "… in heavy clay soil, …" was changed to "… in clay loam soil, …" .

**3)Comment**

Start on 299. Line 307. Reference conceptual model at line 307.

***Response***

In line 307 (older annotated version) we could not find "… conceptual model …".

**4)Comment**

Clarify "soil moisture" in line 308

**Response**

In lines 274 to 275, we revised the text to "… soil moisture (including both saturated and unsaturated soil moisture) … "

**5)Comment**

Line 321 subheading. Change to water table 'elevation' instead of 'position.'

**Response**

The change has been made (line 288).

**6)Comment**

Line 325- Figure 2 not 2a

**Response**

It was revised (Line 292).

**7)Comment**

Line 326. Change water level elevation to "water table elevation"

Aren't all layers "semi-permeable" as defined by their Ks?

**Response**

"… water level elevation …" was changed to "… water table elevation …" (Line 291). You are right, all layers can be semi-permeable based on their Ks, but in this study we assumed the layer at the bottom of the soil layer as semi-permeable layer.

**8)Comment**

Line 333. field capacity defined with different symbols (thetafc vs. hfc)

Do you mean volumetric water contents at fc and matric potential at fc?

**Response**

They both refer to field capacity, one expressed in volume fraction ($\theta_{fc}$ mean volumetric water content) and the other expressed in units of water column ($h_{fc}$). We clarified this in line 305.

**9)Comment**

Line 342. Matric head or pressure head? Matric implies suction when the soil is dry and under tension. Matric potential ~ 0 at saturation.

*Response*

In lines 311 and 313 we changed the "matric head" to "water table".

**10)Comment**

Line 345. You switched to water table level in #3. Revise 1 and 2 to be consistent with #3.

*Response*

Revised to "water table".

**11)Comment**

Line 350-351. Much of the water between theta-fc and theta-wp is held in micropores and would not likely drain from gravitational forces and would only be available to plant roots.

*Response*

Based on the generic soil characteristic curve illustrated in Fig. 2, when the water table drops further than capillary fringe thickness (Condition 3), a part of the soil layer above the water table is occupied by capillary fringe with moisture content equal to $\theta_C$ (as we defined in the new revision) and above this layer is a layer of soil with soil moisture equal to field capacity. In Condition 2 the soil moisture in this upper layer is equal to $\theta_C$, and after changing from Condition 2 to Condition 3, the moisture content in this upper layer gradually shifts from $\theta_C$ to field capacity.  CRHM soil layers have an empirical soil moisture depletion curve taken from Zahner (1967) for soil moisture withdrawals to support transpiration.

**12)Comment**

Line 360. Hooghoudt's equation does not use matric potential it uses water table depth.

*Response*

The reviewer is correct. We've used the water table. This has been corrected (Line 338) "... matric potential" changed to "... water table elevation above the tile pipe". We apologize for the oversight.

**13)Comment**

Line 369-370. "the case with our case study…" Revise

***Response***

Line 340: ''In the particular case of the case study site, …'' was revised to '' At the study site, …''.

**14)Comment**

Line 389. Define $\varphi c$ again.

***Response***

In line 355 and 356 we added $\varphi_c$ to the relevant definition for this parameter as '' … , which consists of drainable water in the soil ($\varphi_c$) …''

**15)Comment**

Line 394. Matric potential or water table depth?

***Response***

Line 364: we changed '' … matric potential:'' to ''… water table elevation:''

**16)Comment**

Line 396. $\varphi c$ not defined previously. You should consider defining all variables in eq. 3

***Response***

In Line 356 $\varphi_c$ was defined, before Equations 2 and 3.

**17)Comment**

Lines 422-424. Sounds like you are implying that the elevated matric potential is causing the increase in soil moisture?

***Response***

Line391 to 395: Since the matric potential is a negative number, we meant that during soil water seepage and concomitant soil moisture decrease, the matric potential becomes more negative, i.e., negative matric potentials with larger absolute values.

**18)Comment**

Line 429. How is Dd determined?

***Response***

We have added a new Appendix D, in that we discuss how Dd (time delay) was assessed.

**19)Comment**

Line 446. Include units for field capacity

***Response***

Lines 418-419: we added the unit of "… (volumetric water content) …".

**20)Comment**

Line 451. Matric potential or WTD. Believe the latter. What's the difference between dependent and state variables?

***Response***

Line 424: here it can be both matric potential or water table depth (WTD), since they are related. State variables are the variables that can change freely and independently during the model run (i.e. soil moisture content), which mostly represent the storage states. While the dependent variables are those who are related to other parameters such as the water table (WT) which in this study is calculated based on soil moisture and soil porosity.

In lines 284-285: "State variables" were changed to "variables".

In Line 423-424: "state variables" was changed to "outflows".

**21)Comment**

Line 452-454. Revise this to include a statement explaining what you used these methods for (assessing model accuracy)

***Response***

Lines 429 to 433: We have added a couple of sentences to discuss why we used different metrics to assess model performance.

**22)Comment**

Line 492: suggest replacing 'assessed accurately' with 'predicted accurately.' Figure 4. Differentiate panel A and B in the figure caption. Not clear how panel B differs.

***Response***

Line 469: we changed it to "… predicted accurately, …". In Figure 4's caption we added a sentence about the differences between panels a and b.

**23) Comment**

Line 509. Please clarify "observation gaps". How do model statistics characterize fit between observed and predicted flows? Was there consistent bias in model predictions relative to observed flows? Was there seasonal variation in model performance?

**Response**

There were periodic gaps in our field observations due to equipment failure. This has been clarified "Despite the gaps in the observational record dur to periodic equipment failure, the model agrees well with observations.".

We calculated the five performance metrics for soil water table (Table 2), using the observed and predicted water tables where observed values were available. Based on what is presented in Table 2 there was about 10% Bias in model water table predictions. A couple of sentences were added to the end of sections 3.3 (Lines 515-517) and 3.4 (Lines 526-535).

Also, these sentences were added to the end of section 3.4: "We calculated the performance coefficients for May-September and October-April.  The results shows that surface flow biases were large and negative in May-Sept and were smaller and positive during Oct-Apri." The relevant tables for the performance coefficients for May-September and October-April were added to the text.

Table 2. Performance coefficients for surface flow, tile flow and water table (WT/SSS), as well as total (tile + surface) flow, for the simulation period of October 2011 to January 2018. The coefficients were calculated for both hourly and daily flow rates.

| Performance coefficients | Surface flow | Tile flow | WT (SSS) (m) | Total flow | |
|---|---|---|---|---|---|
| NSE[*] | -2.29 | 0.31 | 0.49 | -1.38 | Coefficients calculated for hourly flow rates (mm h⁻¹) |
| RMSE[^] | 0.27 | 0.08 | 0.26 | 0.30 | |
| Bias[#] | 0.54 | 0.24 | 0.14 | 0.28 | |
| PBias[$] | 21.77 | 17.91 | 10.46 | 18.63 | |
| RSR[&] | 1.82 | 0.83 | 0.71 | 1.54 | |
| NSE | -0.73 | 0.29 | 0.50 | 0.01 | Coefficients calculated for daily flow rates (mm d⁻¹) |
| RMSE | 2.04 | 1.72 | 0.24 | 2.92 | |
| Bias | 0.35 | 0.20 | 0.09 | 0.22 | |
| PBias | 35.11 | 19.63 | 9.33 | 21.73 | |
| RSR | 1.31 | 0.84 | 0.70 | 0.99 | |

[*] Nash-Sutcliffe efficiency, [^] Root-Mean-Square Error, [#] Model Bias, [$] Percentage Bias, [&] RMSE-observation standard deviation ratio

**Coefficients for May-September**

(The green color shows the performance coefficient which were improved compared with their original values and the red ones were worsened)

| Performance coefficients | Surface flow | Tile flow | WT (SSS) (m) | Total flow | |
|---|---|---|---|---|---|
| NSE* | -18.98 | 0.19 | 0.40 | -11.76 | Coefficients calculated for hourly flow rates (mm h⁻¹) |
| RMSE^ | 0.26 | 0.03 | 0.12 | 0.26 | |
| Bias# | -1.43 | 0.49 | 0.03 | 0.11 | |
| PBias$ | -142.79 | 48.88 | 3.44 | 10.96 | |
| RSR& | 2.85 | 0.57 | 0.39 | 2.27 | |
| NSE | -3.89 | 0.21 | 0.41 | -1.08 | Coefficients calculated for daily flow rates (mm d⁻¹) |
| RMSE | 1.39 | 0.73 | 0.11 | 1.66 | |
| Bias | -1.43 | 0.49 | 0.02 | 0.11 | |
| PBias | -142.79 | 48.88 | 2.07 | 10.96 | |
| RSR | 1.41 | 0.56 | 0.39 | 0.92 | |

*Nash-Sutcliffe efficiency, ^Root-Mean-Square Error, #Model Bias, $Percentage Bias, &RMSE-observation standard deviation ratio

**Coefficients for October-April**

(The green color shows the performance coefficient which were improved compared with their original values and the red ones were worsened)

| Performance coefficients | Surface flow | Tile flow | WT (SSS) (m) | Total flow | |
|---|---|---|---|---|---|
| NSE* | -0.37 | 0.24 | 0.20 | -0.04 | Coefficients calculated for hourly flow rates (mm h⁻¹) |
| RMSE^ | 0.11 | 0.07 | 0.21 | 0.14 | |
| Bias# | 0.87 | 0.14 | 0.11 | 0.24 | |
| PBias$ | 86.59 | 13.56 | 11.00 | 24.11 | |
| RSR& | 0.90 | 0.67 | 0.77 | 0.79 | |
| NSE | -0.11 | 0.26 | 0.24 | 0.18 | Coefficients calculated for daily flow rates (mm d⁻¹) |
| RMSE | 1.50 | 1.56 | 0.21 | 2.40 | |
| Bias | 0.87 | 0.14 | 0.11 | 0.24 | |
| PBias | 86.59 | 13.56 | 10.58 | 24.11 | |

| RSR | 0.81 | 0.67 | 0.75 | 0.70 |

[*] Nash-Sutcliffe efficiency, [^] Root-Mean-Square Error, [#] Model Bias, [$] Percentage Bias, [&] RMSE-observation standard deviation ratio

**24) Comment**

Figure. 5: The y-axis is a bit confusing. Please clarify the y-axis scale and why it goes negative. Is this implying the water table depth is that far below the soil surface? How could predicted water table be > 0 elevation if 0 = soil surface elevation? Your measured water table depths are never <0 in the figure. Also, it is not clear where the tile depths are located on the figure.

*Response*

This is an issue of using the tile pipe as the datum.  We have added details to the Figure 5 caption: "the water table values on the y-axis are in metres above (or below) the tile pipe". The tile pipe is located at WT=0; thus, the negative values in the figure correspond to the periods when the water table went below the tile pipe.

**25)Comment**

Figure 6. Revise caption. Performance coefficients were not included.

*Response*

This caption has been revised.

**26)Comment**

Line 551. Replace "matric potential levels" with 'water table depth'.

*Response*

Line 523: Since the water table elevations with respect to the tile pipe elevation was reported, we revised it to "… water table elevation …".

**27)Comment**

More explanation of model performance in warranted. What's considered acceptable?

Did performance vary by season?

*Response*

Lines 429 to 433: Further discussion on the acceptable range of performance coefficients were added here. We calculated the seasonal performance coefficients and added them to the text.

**28)Comment**

Line 556. It might be helpful to include the coefficient of determination (R2) along with these.

*Response*

We agree with the reviewer that $R^2$ could be useful, but we consider that the 5 performance metrics used already provide a robust performance assessment. Besides that, the paper is already quite long and, in our view, adding one more metric wouldn't bring a clear added value.

**29)Comment**

Line 567. Field or catchment scale? Catchment implies a larger watershed doesn't it?

*Response*

Line 559: we changed it to "… the field-scale …". But we meant we wanted to find a general value for the capillary fringe thickness that can be used in larger-scale studies.

**30)Comment**

Line 582. How were values normalized? Put on 0 to 1 scale?

*Response*

Lines 567 to 568: Exactly, for example, by dividing the capillary fringe thickness by the tile depth we calculated the normalized thickness of the capillary fringe above the tile. The normalized values are more comparable between different fields and fluctuate around 0 and 1.

**31)Comment**

Lines 572-588. This section is hard to follow and the main point is not clear.

*Response*

In this section (Line 565 to 585) The main purpose of this section is to perform a sensitivity analysis of the effect of drainable water and capillary fringe thickness on tile flow. The variables have been normalized to enable a more generic interpretation and discussion that can be relevant to other case studies. This has been described more clearly in the manuscript (Lines 580-585).

**32)Comment**

Line 598: If water table depth and SSS are synonymous does it makes sense to differentiate?

***Response***

Although these are indeed synonymous, we differentiated WT and SSS because WT represents what we can measure in the field whereas SSS is the saturated soil storage which is calculated as a state variable within the model and is eventually translated to WT to be compared with observed WTs.

**33)Comment**

Line 604. "capillary fridge" to fringe. Also clarify tile discharge

***Response***

Line 596: "…fridge …" was revised to "…fringe …".

Line 596: By "discharge" we meant the discharge of the soil moisture to groundwater through the semi-permeable layer at the bottom pf the soil layer. We changed the "discharge" to "percolation".

**34)Comment**

Line 606: Matric potential or water table depth?

***Response***

Line 598: we changed it to "While the water table fluctuates …".

**35)Comment**

Line 608. Replace matric potential with water table depth.

***Response***

Lines 600: It was revised.

**36)Comment**

Line 610-613. Suggest further clarifying results from Figure 9 in discussion.
***Response***

Lines (600-608): We have clarified the results

**37)Comment**

Line 553-554. Include more discussion on model performance for surface runoff.

*Response*

Line 526 to 535: We added a sentence explaining why the model performance in simulation of surface flow is weaker.

**38)Comment**

Line 624-634. Not sure this section is necessary.

*Response*

Line 623 to 632: The paragraph was deleted.

**39)Comment**

Line 638. Replace matric potential with water table depth.

*Response*

Line 636: It was revised.

**40)Comment**

Line 642: How did you determine they were "equally important"?

*Response*

Line 640: we revised it to "…  are important to account for …".

**41)Comment**

Line 657. Clarify 'holding capacity'. Soil water availability?

*Response*

Line 655: we have changed "holding capacity" to "non-drainable water"

**42)Comment**

Line 670. Writing "WT/SSS" makes it seem like a ratio

*Response*

Line 668: It was changed to " … WT …".

**43)Comment**

Line 676. Clarify water level in last portion of the sentence.

*Response*

Line 673: It was revised to "… water table …".

**44)Comment**

Line 695. follow up here is needed to clarify your point and link to the capillary fringe.

*Response*

Line 693 to 696: A brief follow up discussion was added here.

**45)Comment**

Lines 699-715. Add more on how this relates to a dynamic capillary fringe.

*Response*

Line 700-715: In this section we want to show the effect of regional groundwater fluctuation on the tile flow and water table fluctuations ( and correspondingly the dynamic of capillary fringe). We can see that the general fluctuation of soil water table depends on the seasonal pattern of groundwater fluctuations.

**46)Comment**

Line 723. Change rain 'drops' to rain events.

*Response*

Line 723: It was "rapid drops in observed WT …", so, we kept it without change.

**47)Comment**

Line 730. Add a concluding sentence or two to pull together the overall importance of regional groundwater dynamics on tile flows.

*Response*

Lines 729 to 732: Thank you for the suggestion, which we agree. We've added a concluding sentence.

**48)Comment**

Line 743. Replace 'matric potential' with water table depth.

*Response*

Line 742-743: It was revised to " … water table elevation …", as the water table were reported as the elevation from the tile pipe.

**49)Comment**

Line 746. Did you measure matric potential?

*Response*

We did not measure matric potential as a continuous time series, but we have the sparse measurements of the matric potential in this field by our team members for other studies. We did not have the continuous time series of the matric potential, but we know that it continuously changes with variations in the soil water table elevation.

**50)Comment**

Line 768. Your site was clay loam soil

*Response*

Line 768: it was revised.

Other items

**51)Comment**

How does the model handle impact of frozen soil on tile flow?

*Response*

We did not consider the impact of frozen soil on tile flow in this study, but some soil modules in CRHM are capable of predicting the effect of frozen soil on soil moisture (Pomeroy et al., 2007; 2022).

**52)Comment**

Hooghoudt assumes no surface ponding, but your site experienced some ponding, as mentioned in the results. How might this affect water table/flow predictions?

*Response*

As we discussed in section 2.4.3, since the surface ponding happened very rarely and for short periods, we did not implement the version of Hooghoudt's equation with surface ponding in the module. But in future version we will add this capability.

**53)Comment**

Reread and revise Appendices for grammar as needed

***Response***

The writing was checked and revised in appendices.

**Referee #2**

**Suggestions for revision or reasons for rejection**

The authors have done a great job at addressing most of the comments from the reviewers and I have no comments to changes made. I also acknowledge that the study is a preliminary model development based on a single site. However, there are two elements which I still do not find quite satisfactory, these regard the parametrization and application to other locations. Please see my specific comments below which refer to the previous comment numbers. I feel these comments would be relatively straight forward to address.

**1)Comment**

Comment 6 by Reviewer 2

You would still need a sensitivity analysis to claim that the capillary fringe drainable water and the capillary fringe thickness are the most important parameters. How can you state that they are the most important without a sensitivity analysis and how important are the other parameters? Several review comments address the parametrization of the module also outside the particular study site, and it is important to understand the importance of the many parameters. Which can be set to global values, which need detailed investigation or calibration. Table 1, or an additional table, should include sensitivities to the simulated drainflow, and ideally indications of how there parameter values could be obtained. With so many parameters there needs to be some guidance on which parameters to focus on and why.

***Response***

A new sensitivity analysis that shows the sensitivity of cumulative tile flow, average water level elevation and cumulative soil to groundwater outflow to six parameters including soil saturated hydraulic

conductivity, soil thickness, capillary fringe thickness, capillary fringe drainable water, sine function amplitude and sine function intercept was added as Appendix D.

Also, in Appendix D we added a section about how to evaluate the parameters for a new site.

**2)Comment**

Comment 7,8 and 9, by Reviewer 2

I understand that this is a preliminary study and you have focused on a single site and tried to learn from that specific site. However, the paper needs a section that specifically addresses how this module would be applied and parameterized at a location without any water table or drainflow observations. Without considering this step it is very hard to understand the value of the model. In modelling we are interested in estimating variables where they are not already measured. Such a section should refer to a sensitivity analysis and discuss the key parameters and how they would be obtained at other locations, also the need for adjusting groundwater levels annually, and how that information would be attained, should be discussed. And the section would be even better, if you could validate the module with just a single additional site, where drainflow is measured, but where you would need to transfer generic parameters from your study site or use the Deduction, Induction, Abduction approach.

***Response***

As we discussed in our response to previous comment, we have added a section in Appendix D about how to assess the TDM parameters for a new site, probably the site without any water table or drainflow observation. Also, in this section in Appendix D we discuss specifically how we could assess the parameters for the sine function for new sites. Unfortunately, it is not possible for us to validate the model by using it at a new site due to lack of proper data. But we certainly plan to implement the new TDM model to new sites in the future and continue improving it.

---

## Author Response (AR3)

**Responses to the comments from Referee #1 regarding manuscript number egusphere-2023-142, entitled "*Developing a tile drainage module for Cold Regions Hydrological Model: Lessons from a farm in Southern Ontario, Canada*", authors: Mazda Kompanizare et al.**

The numbers on our responses to the referee's comments are based on the latest annotated version of the manuscript.

General comments

The authors effectively addressed many of the reviewer comments in their revision to improve the manuscript's clarity, however considerable revision is still needed before publication. Please refer to line-by-line comments that follow.

Abstract

**1)Comment**

Line 12. "…extensively in [poorly drained] agricultural lands…" Suggest adding 'poorly drained'.

***Response***

Line 13. It was added.

**2)Comment**

Line 16. Clarify 'runoff'. Surface, subsurface, both?

***Response***

Line 17. It was changed to "… agricultural surface and subsurface runoff."

**3)Comment**

Line 26. "Shorted" should be 'shorter'

***Response***

Line 28. It was corrected.

**4)Comment**

Line 25-29. Consider condensing this sentence, a bit hard to follow

*Response*

Line 27-31: The sentence was shortened.

Introduction

**5)Comment**

The introduction is still long and could be more concise.

*Response*

This section has been condensed.

**6)Comment**

Lines 45-50. Lowering the seasonally high water table in poorly drained fields is the main function of tiles drains

*Response*

Line 50. We added "…lower the seasonally high water table in poorly drained fields, …" to the sentence.

**7)Comment**

Line 61. "…that can represent tile drainage". Consider being more descriptive re: tile drainage…what type of flows, matrix? Gravitational? What about surface runoff?

*Response*

Line 66-71: The sentence was revised and some more details were added.

**8)Comment**

Line 76. "Since the use of tile drainage has become popular… ". Hasn't tile drainage been used extensively for decades? Do you have any recent tiel adoption trend data specific to Canada that you can cite?

*Response*

As presented in the study by Kokulan (2019) "Although tile drainage has not historically been used in Canadian Prairies, an increasing frequency of multiday spring and summer storms in these regions (Shook and Pomeroy, 2012) has caused farmers in provinces such as Manitoba and Saskatchewan to

install tile drains at an accelerated rate to tackle the unprecedented waterlogging conditions in their crop fields (Cordeiro and Ranjan, 2012; Kokulan et al., 2019a)." Tile drain usage is also increasing in the Great Lakes region (OMAFRA, 2023; https://geohub.lio.gov.on.ca/datasets/ontarioca11::tile-drainage-area/explore?showTable=true ), particularly in fields that are imperfectly drained such as the study field.

Lines 83-84: The relevant references were added.

Lines 1052-1054 and 1106-1108: Two references were added to the reference list.

**9)Comment**

Line 100. You mention catchment scale but your study is at the field scale, please clarify whether the aim of CHRM-TD is catchment or field scale.

*Response*

We thank the reviewer for their comment. In this sentence, we were referring to the PDF-based numerical models, which can be used in both field and catchment scales and not specifically about CRHM-TD. In response to the earlier suggestion to shorten the introductory section, we have removed this paragraph.

**10)Comment**

Lines 104-108. What about the hydraulic gradient?

*Response*

Line 112-113: we added "…hydraulic gradient …" to the sentence.

**11)Comment**

Line 110. Suggest changing "Many" to 'Some studies'. Also some of the references cited are quite outdated.

*Response*

Line 119: "Many …" was removed. The references mentioned here are some of the first ones considering drainable water in tile flow calculations.

**12)Comment**

Line 140. Integrate the last sentence with the previous one.

*Response*

Lines 145-149: The two sentences have been integrated.

**13)Comment**

Line 154-155. "…which are increasingly being artificially drained". Citation?

***Response***

Lines 164-165: Three references have been added.

**14)Comment**

Line 162. Soil type or series, not texture.

***Response***

Line 172: It was revised to " Soil type …".

**15)Comment**

Line 192. If snowmelt processes were accounted for they should be better explained. How was melting estimated? What about infiltration with partially frozen soils?

***Response***

This section has been rewritten. Our previous submission included a section that described the capabilities of the CRHM platform and later mentioned the modules used to create a CRHM model for this study. This has led to confusion, so we have removed references to modules not directly deployed in the current study, and we have referred the reader to Pomeroy et al. (2022) for a more comprehensive description of CRHM's capabilities, including those within and beyond the scope of this study.

**16)Comment**

Line 197. "Water quality" needs further clarification. What specific nutrients? Sediment?

***Response***

Lines 209-210: The sentence was deleted.

**17)Comment**

Line 223. Preferential flow is likely an important mechanism at your site since the texture is a clay loam. Preferential flow even occurs readily in silt loams. Maybe just state that you did not model it for this study and will be assessed in future studies?

***Response***

We agree with the reviewer that preferential flow can be highly important in both clay loams and silt loams. However, we used hydrograph analyses (Macrae et al., 2019) and conservative tracers (electrical conductivity and major ions, as well as temperature) over multiple years (Pluer et al., 2020) and found minimal preferential flow at this site as well as other similar sites. For this reason, preferential flow was not included in this study. However, we will certainly continue exploring this transport mechanism in future studies. This statement is retained in the revised manuscript (Lines 238-241).

**18)Comment**

Line 227. Our research shows that soil freezing still happens with snow cover, with depth and extent depending on snowpack depth and other radiative factors.

*Response*

We agree and have modified our sentence to reflect this. We have improved our justification of why we chose to exclude freeze-thaw here. "Freeze-thaw of soil can occur in the study region, leading to partially frozen soils. However, the extent of freezing can differ with snowpack development and other radiative factors. Data collected over an 8-year period at this site found soil freezing was restricted to brief periods and such freezing never extended below 10 cm depth. Such shallow depth of freezing does not meet the criteria for frozen soil infiltration where the influence of ice in soil pores must be considered in soil water movement calculations (Zhao and Gray, 1999). Consequently, freeze-thaw processes were not deemed critical for representation in our modelling study, though they are a CRHM platform capability and could easily be added should frozen soils occur."

**19)Comment**

Line 241. How good was the regression relationship for the rating curve?

*Response*

Regression relationships were $R^2 > ~ 0.5$. This is because during high-flow periods, water levels would rise in our access pipes due to impeded flow downstream. However, the depth-velocity sensor was largely functioning during such periods. This has been mentioned in the paper.

**20) Comment**

Line 245. You mention "forcing" with other covariates but do not present or discuss forcing data. I suggest not using this term unless you did use it to force the model.

*Response*

We used air temperature, wind speed, relative humidity, incoming solar radiation and precipitation, to assess the amount of evapotranspiration as well as surface runoff.

Line 274: We changed the "… used to force …" to "… were implemented in …".

**21)Comment**

Line 344. Was Ks measured or assumed?

*Response*

Ks was estimated during model calibration.

**22)Comment**

Line 347. "…was used to estimate"

*Response*

Line 375. It was corrected.

**23)Comment**

Line 376. "…into the sire from adjacent farms". Replace 'farms' with 'fields'

*Response*

Line 404: It was replaced.

**24)Comment**

Line 386. Clarify "This approach" at the start of the sentence.

*Response*

Lines 414-415: It was added " …, using the sine function, …" to clarify "This approach, …"

**25)Comment**

Line 389-392. This sentence is long and hard to follow, suggest revising.

*Response*

Lines 417-421: The sentence is rewritten.

**26)Comment**

Line 421. State somewhere that these methods were used to assess model accuracy

***Response***

Line 454: "These methods were used to assess model accuracy" was added.

**27)Comment**

Table 1. Remove "Source" as a column heading if it is not used.

***Response***

It was removed.

**28)Comment**

Line 452. Suggest revising to "…the near absence of flow"

***Response***

Lines 489-490: It was revised.

**29)Comment**

Line 462-463. Suggest revising to" Although peak tile drainage flow was not always…"

***Response***

Line 499-500: It was rewritten as it was suggested.

**30)Comment**

Line 474. Saturated soil storage and water table depth appear to be used interchangeably, which causes some confusion. Suggest sticking with one term or the other if you are implying the same physical state or clarifying the use of both terms.

***Response***

To prevent confusions, in Figure 5, and the vertical axis title was changed to SS. Also, in line 512 we removed "…observed water table …".

**31)Comment**

Figure 5. Same comment as above. Y-axis lists 'SSS' and 'WT'- suggest sticking with one or the other as per above comment.

***Response***

Vertical axis title in Figure 5 was changed to saturated storage (SS).

**32)Comment**

Lines 506-509. Provide more specifics about the "systematic issues" and provide some ideas on why the surface flow is not predicted well and how you plant to improve it.

*Response*

As presented in lines 536 -540, one of the systematic issues in the surface flow simulation in CRHM is that CRHM adds and removes water instantaneously to depressional storage and so was not able to calculate the lag-time related to filling up the ponded areas and outflow from the ponds which is proportional to the water level within ponded areas. By adding those storage related lag-times and route to surface runoff the simulated surface flows were closer to measured.

**33)Comment**

Lines 513-526. Suggest providing additional context here. What are these collective differences suggesting about the overall model?

*Response*

Lines 551-553 and 567-570: Some additional context was added and some sentences about the overall performance of the model were added in section 3.4.

**34)Comment**

Figure 6. Might be helpful to present the R2 values for relationships.

*Response*

To be consistent with Table 2 we added NSE, RMSE, Bias, PBias and RSR to the cumulative surface and total flows in Figure 6.

**35)Comment**

Line 545. Change "have" to 'had' a strong influence"

*Response*

Line 588: It was changed to "had".

**36)Comment**

Line 546. "…that flowed into tiles.

***Response***

Line 590: It was revised.

**37)Comment**

Table 2. Revise for consistent significant digits across the table

***Response***

Table has been revised to have consistent two digits after the decimal point.

**38)Comment**

Line 591. "ore" should be 'more'

***Response***

Line 636: It was corrected.

**39)Comment**

Line 598. Again, suggest sticking with either water table (WT) or saturated soil storage (SSS) but not using them interchangeably to avoid unnecessary confusion.

***Response***

Lines 637-638. We have corrected this to SS.  Also, in Figure 8 the water Table (WT) observations were presented we deleted "water table" and now presented this as "saturated storage" because, water table observations and the simulated soil saturated storage (SS) are equivalent.  In Figure 8 vertical and horizontal axis and the figure caption also were revised.

**40)Comment**

Figure 9 caption. Same comment as above- use either SSS or WT but not both. Also, should it be water table depth or just water table?

***Response***

We now show values as Saturated Storage in Figure 9.

**41)Comment**

Line 612-624. Can you use the relationship you found between capillary fringe and drainable water to improve observed vs. predicted flows? While this might be too much to add to your results, discussing how one would use these data to improve CRHM-TD module seems like an important area to discuss.

*Response*

The relationship between capillary fringe thickness and drainable water, as well as groundwater fluctuations to tile flow, cannot be summarized in a simple relationship or equation, and to use these relationships, one should use the CRHM-TD module.

Another important point is that some of these relationships are more controlled by the existing delay in the drainage of water from the capillary zone. In our future research we will work on finding some simple approach to consider this delay in our simulations.

**42)Comment**

Line 627. Should 'K' be "Ks" for saturated hydraulic conductivity?

*Response*

Yes, it should. Also in Table 1, K had been defined as saturated hydraulic conductivity, based on the referee's suggestion, we have changed K to Ks in the whole document (i.e. in Line 674 and in Table 1).

**43)Comment**

Line 635. Same comment re: WT or SSS- use one or the other or explain reasoning for using both terms.

*Response*

Line 682 and Line 693. It was changed it to SS.

**44)Comment**

Line 648. Same as above.

*Response*

Line 696. It was changed to SS.

**45)Comment**

Line 666-667. Similar comment as above- can you use the new relationships to improve flow predictions? How would this process unfold if you are not able to apply it now?

*Response*

In developing TDM, we wanted to test and capture the effect of capillary fringe thickness, drainable water, and saturated storage fluctuations on tile flow rates. Section 2.4 of the paper is explains how these three control factors are represented in the new module, which can be easily replicated and benefit the broader modelling community. Specifically, Fig. 2 shows how the representation of the capillary fridge thickness was divided into three phases, which were implemented through "if, then" statements in the model development. Equations 1 to 4 show how tile flow was calculated, as well as the effect of soil moisture and water table.

**46)Comment**

Line 669-670. What about the fact that tiles are 1 m deep and soil moisture was measured at 0.5 m?

*Response*

Lines 714-717. Those soil moisture measurements were conducted during 2011 to 2014 and were related to other studies in this area. We just used them here as additional observations. We would ideally have wanted soil moisture observations up to the depth of tile pipe (1 m), but even the moisture content observations at the depth of 0.5 m showed that almost 90% of the gravitational soil moisture drains out within 0.5 to 2.5 h.

**47)Comment**

Lines 681-692. Shouldn't the role of evapotranspiration be included here?

*Response*

Lines 729 to 737: Two sentences were added about the role of evapotranspiration.

**48)Comment**

Line 721. Delete space between sentences.

*Response*

It was fixed.

**49)Comment**

Line 742. As previously mentioned, preferential flow is likely an important mechanism in your field given the texture is clay loam.

*Response*

Future developments can explore this but our current studies have not shown it to be substantial at this site. Please check our response to comment #17, as well.

**References**

Cordeiro M.R.C. Ranjan R.S. (2012) "Corn yield response to drainage and subirrigation in the Canadian Prairies" Transactions of the ASABE, 55(5), 1771-1780.

Kokulan V. (2019) "Environmental and Economic Consequences of Tile Drainage Systems in Canada" The Canadian Agri-Food Policy Institute, CAPI.

Kokulan V., Macrae M.L., Ali G.A., Lobb D.A. (2019a) "Hydroclimatic controls on runoff activation in a artificially drained, near-level vertisolic clay landscape in a Prairie climate" Hydrological Processes, 33:602-615. DOI:10.1002/hyp.13347

Shook K., Pomeroy, J. (2012) "Changing in the hydrological character of rainfall on the Canadian prairies" Hydrological Processes, 26(12), 1752-1766.

Zhao L., Gray D.M. (1999) "Estimating snowmelt infiltration into frozen soils" Hydrol. Process. 13 (12-13_, 1827-1842.